# Graph Neural Network Generalization
# with Gaussian Mixture Model Based Augmentation

**Yassine Abbahaddou** [1] [*]   **Fragkiskos D. Malliaros** [2]   **Johannes F. Lutzeyer** [1]
**Amine M. Aboussalah** [3]   **Michalis Vazirgiannis** [1] [4]

## Abstract

Graph Neural Networks (GNNs) have shown great promise in tasks like node and graph classification, but they often struggle to generalize, particularly to unseen or out-of-distribution (OOD) data. These challenges are exacerbated when training data is limited in size or diversity. To address these issues, we introduce a theoretical framework using Rademacher complexity to compute a regret bound on the generalization error and then characterize the effect of data augmentation. This framework informs the design of GRATIN, an efficient graph data augmentation algorithm leveraging the capability of Gaussian Mixture Models (GMMs) to approximate any distribution. Our approach not only outperforms existing augmentation techniques in terms of generalization but also offers improved time complexity, making it highly suitable for real-world applications. Our code is publicly available at: https://github.com/abbahaddou/GRATIN.

## 1. Introduction

Graphs are a fundamental and ubiquitous structure for modeling complex relationships and interactions. In biology, graphs are employed to represent complex networks of protein interactions and in drug discovery by modeling molecular relationships (Gaudelet et al., 2021; Jagtap et al., 2022). Similarly, social networks capture relationships and community interactions (Aboussalah et al., 2023a; Zeng et al., 2022; Malliaros & Vazirgiannis, 2013; Newman et al., 2002). To address the unique challenges posed by graph-structured data, GNNs have been developed as a specialized class of neural networks designed to operate directly on graphs. Unlike traditional neural networks that are optimized for grid-like data, such as images or sequences, GNNs are engineered to process and learn from the relational information embedded in graph structures. GNNs have demonstrated state-of-the-art performance across a range of graph representation learning tasks such as node and graph classification, proving their effectiveness in various real-world applications (Vignac et al., 2023; Corso et al., 2023; Duval et al., 2023; Castro-Correa et al., 2024; Chi et al., 2022; Panagopoulos et al., 2024; Aboussalah & Ed-dib, 2025).

Despite their impressive capabilities, GNNs face significant challenges related to generalization, particularly when handling unseen or out-of-distribution (OOD) data (Guo et al., 2024; Li et al., 2022). OOD graphs are those that differ significantly from the training data in terms of graph structure, node features, or edge types, making it difficult for GNNs to adapt and perform well on such data. This challenge is also faced when GNNs are trained on small datasets, where the limited data diversity hampers the model's ability to generalize effectively. To address these challenges, the community has explored various strategies to improve the robustness and generalization ability of GNNs (Abbahaddou et al., 2024; Yang et al., 2023).

Generalization bounds for GNNs have been derived using various theoretical tools, such as the Vapnik-Chervonenkis (VC) dimension (Pfaff et al., 2021; Garg et al., 2020) and Rademacher complexity (Yin et al., 2019; Esser et al., 2021). Furthermore, Liao et al. (2021) were among the first to establish generalization bounds for GNNs using the PAC-Bayesian approach. Neural Tangent Kernels have also been employed to study the generalization properties of infinitely wide GNNs trained via gradient descent (Jacot et al., 2018; Du et al., 2019; Huang et al., 2024). While most existing research has focused on the node classification task, enhancing generalization in graph classification presents unique challenges. Techniques to improve generalization in graph classification can be broadly categorized into architectural and dataset-based strategies (Tang & Liu, 2023; Buffelli et al., 2022). On the dataset side, techniques like adversar-

---

[*]Work partially completed during a research visit to NYU. [1]LIX, École Polytechnique, IP Paris [2]Université Paris-Saclay, CentraleSupélec, Inria [3]NYU Tandon School of Engineering [4]MBZUAI. Correspondence to: Yassine Abbahaddou <yassine.abbahaddou@polytechnique.edu>.

*Proceedings of the $42^{nd}$ International Conference on Machine Learning*, Vancouver, Canada. PMLR 267, 2025. Copyright 2025 by the author(s).

ial training and data augmentation play a significant role. More broadly, data augmentation methods create synthetic or modified graph instances to enrich the training set, reducing overfitting and enhancing the model's adaptability to diverse graph structures. Data augmentation has shown its benefits across different types of data structures such as images (Krizhevsky et al., 2012) and time series (Aboussalah et al., 2023b). For graph data structures, generating augmented versions of the original graphs, such as by adding or removing nodes and edges or perturbing node features (Rong et al., 2020; You et al., 2020), allows for the creation of a more varied training set. Inspired by the success of the Mixup technique in computer vision (Rebuffi et al., 2021; Dabouei et al., 2021; Hong et al., 2021), additional methods such as $\mathcal{G}$-Mixup and GeoMix have been developed to adapt Mixup for graph data (Ling et al., 2023; Han et al., 2022). These techniques combine different graphs to create new, synthetic training examples, further enriching the dataset and enhancing the GNN's ability to generalize to new unseen graph structures.

In this work, we introduce GRATIN, a graph augmentation technique based on Gaussian Mixture Models (GMMs), which operates at the level of the final hidden representations. Specifically, guided by our theoretical results, we apply the Expectation-Maximization (EM) algorithm to train a GMM on the graph representations. We then use this GMM to generate new augmented graph representations through sampling, enhancing the diversity of the training data.

The contributions of our work are as follows:

- **Theoretical framework for generalization in GNNs.** We introduce a theoretical framework that allows us to rigorously analyze how graph data augmentation impacts the generalization GNNs. This framework offers new insights into the underlying mechanisms that drive performance improvements through augmentation.

- **Efficient graph data augmentation via GMMs.** We propose GRATIN, a fast and efficient graph data augmentation technique leveraging GMMs. This approach enhances the diversity of training data while maintaining computational simplicity, making it scalable for large graph datasets.

- **Comprehensive theoretical analysis using influence functions.** We perform an in-depth theoretical analysis of our augmentation strategy through the lens of influence functions, providing a principled understanding of the approach's impact on generalization performance.

- **Empirical Validation.** Through experiments on real-world datasets we confirm GRATIN to be a fast, high-performing graph augmentation scheme in practice.

## 2. Background and Related Work

**Notation.** Let $\mathcal{G} = (\mathcal{V}, \mathcal{E})$ denote a graph, where $\mathcal{V}$ represents the set of vertices and $\mathcal{E}$ represents the set of edges. We use $p = |\mathcal{V}|$ to denote the number of nodes. For a node $v \in V$, let $\mathcal{N}(v)$ be the set of its neighbors, defined as $\mathcal{N}(v) = \{u \colon (v, u) \in \mathcal{E}\}$. The degree of vertex $v$ is the number of neighbors it has, which is $deg(v) = |\mathcal{N}(v)|$. A graph is commonly represented by its adjacency matrix $\mathbf{A} \in \mathbb{R}^{p \times p}$, where the $(i, j)$-th element of this matrix is equal to the weight of the edge between the $i$-th and $j$-th node of the graph and a weight of zero in case the edge does not exist. Additionally, in some cases, nodes may have associated feature vectors. We denote these node features by $\mathbf{X} \in \mathbb{R}^{p \times d}$ where $d$ is the dimensionality of the features.

**Graph Neural Networks (GNNs).** A GNN model consists of multiple neighborhood aggregation layers that use the graph structure and the feature vectors from the previous layer to generate updated representations for the nodes. Specifically, GNNs update a node's feature vector by aggregating information from its local neighborhood. Consider a GNN model with $T$ neighborhood aggregation layers. Let $\mathbf{h}_v^{(0)}$ denote the initial feature vector of node $v$, which is the corresponding row in $\mathbf{X}$. At each layer $t > 0$, the hidden state $\mathbf{h}_v^{(t)}$ of node $v$ is updated as follows:

$$\mathbf{m}_v^{(t)} = \text{AGGREGATE}^{(t)}\Big(\big\{\mathbf{h}_u^{(t-1)} \colon u \in \mathcal{N}(v)\big\}\Big),$$

$$\mathbf{h}_v^{(t)} = \text{COMBINE}^{(t)}\Big(\mathbf{h}_v^{(t-1)}, \mathbf{m}_v^{(t)}\Big),$$

where AGGREGATE$(\cdot)$ is a permutation-invariant function that combines the feature vectors of $v$'s neighbors into an aggregated vector. This aggregated vector, together with the previous feature vector $\mathbf{h}_v^{(t-1)}$, is fed to the COMBINE$(\cdot)$ function, which merges these two vectors to produce the updated feature vector of $v$.

After $T$ iterations of neighborhood aggregation, a GNN typically produces a graph-level representation by first applying a permutation-invariant readout function, e.g., a *Sum* operator, to the final node embeddings. This aggregated output is subsequently passed through a trainable neural network, such as a multi-layer perceptron (MLP), denoted by $\Psi$ and referred to as the *post-readout* neural network, to produce the final graph-level predictions or representations. Mathematically, this process can be expressed as

$$\mathbf{h}_{\mathcal{G}} = \Psi \circ \text{READOUT}\Big(\big\{\mathbf{h}_v^{(T)} \colon v \in \mathcal{V}\big\}\Big).$$

Two popular GNN architectures are Graph Convolution Networks (GCN) and Graph Isomorphism Networks (GIN) (Kipf & Welling, 2017; Xu et al., 2019). The exact expression of these models can be found in Appendix C.

**Data Augmentation for Graphs.** Graph data augmentation has become essential to enhance the performance and robust-

ness of GNNs. Classical graph augmentation techniques focus on structural modifications to generate augmented graphs. Key methods here include DropEdge, DropNode, and Subgraph sampling (Rong et al., 2020; You et al., 2020). For instance, DropEdge randomly removes a subset of edges from the graph during training, improving the model's robustness to missing or noisy connections. Similarly, DropNode removes certain nodes as well as their connections, assuming that the missing part of nodes will not affect the semantic meaning, i.e., the structural and relational information of the original graph. Subgraph sampling, on the other hand, samples a subgraph from the original graph using random walks to use as a training graph.

Beyond classical methods, recent advancements have explored more sophisticated augmentation techniques, focusing on manipulating graph embeddings and leveraging the geometric properties of graphs. Following the effectiveness of the Mixup technique in computer vision (Rebuffi et al., 2021; Dabouei et al., 2021; Hong et al., 2021), several works describe variations of the Mixup for graphs. For example, the Manifold-Mixup model conducts a Mixup operation for graph classification in the embedding space. This technique interpolates between graph-level embeddings after the READOUT function, blending different graphs in the embedding space (Wang et al., 2021). Similarly, $\mathcal{G}$-Mixup (Han et al., 2022) uses graphons to model the topological structures of each graph class and then interpolates the graphons of different classes, subsequently generating synthetic graphs by sampling from mixed graphons across different classes. It is important to note that $\mathcal{G}$-Mixup operates under a significant assumption: graphs belonging to the same class can be produced by a single graphon. Other advanced techniques include $S$-Mixup method, which interpolates graphs by first determining node-level correspondences between a pair of graphs (Ling et al., 2023), and FGW-Mixup, which adopts the Fused Gromov-Wasserstein barycenter to compute mixup graphs but suffers from heavy computation time (Ma et al., 2024). Finally, GeoMix (Zeng et al., 2024) leverages Gromov-Wasserstein geodesics to interpolate graphs more efficiently. By leveraging these structural augmentation techniques, GNNs can better generalize to unseen graph structures.

## 3. GRATIN: Gaussian Mixture Model for Graph Data Augmentation

In this section, we introduce the mathematical framework for graph data augmentation and its connection to the generalization of GNNs. Then, we present our proposed model GRATIN, which is based on GMMs for graph augmentation.

### 3.1. Formalism of Graph Data Augmentation

We focus on the task of graph classification, where the objective is to classify graphs into predefined categories. Let $\mathcal{D}$ denote the distribution of graphs. Given a training set of graphs $\mathcal{D}_{\text{train}} = \{(\mathcal{G}_n, y_n) \mid n = 1, \ldots, N\}$, $\mathcal{G}_n$ is the $n$-th graph and $y_n$ is its corresponding label belonging to a set $\{0, \ldots, C\}$. Each graph $\mathcal{G}_n$ is represented as a tuple $(\mathcal{V}_n, \mathcal{E}_n, \mathbf{X}_n)$, where $\mathcal{V}_n$ denotes the set of nodes with cardinality $p_n = |\mathcal{V}_n|$, $\mathcal{E}_n \subseteq \mathcal{V}_n \times \mathcal{V}_n$ is the set of edges, and $\mathbf{X}_n \in \mathbb{R}^{p_n \times d}$ is the node feature matrix of dimension $d$. The objective is to train a GNN $f(\cdot, \theta)$ that can accurately predict the class labels for unseen graphs in the test set $\mathcal{D}_{\text{test}} = \{\mathcal{G}_n^{\text{test}} \mid n = 1, \ldots, N_{\text{test}}\}$. The classical training approach involves minimizing the following loss function,

$$\mathcal{L} = \sum_{n=1}^{N} \ell(f(\mathcal{G}_n, \theta), y_n), \tag{1}$$

where $\ell$ denotes the cross-entropy loss function.

To improve the generalization performance of GNNs, we introduce a graph data augmentation strategy. For each training graph $\mathcal{G}_n$ in the dataset, we generate $M$ augmented graphs, denoted as $\{\widetilde{\mathcal{G}}_{n,m}, \widetilde{y}_{n,m} \mid m = 1, \ldots, M\}$, where $M$ is the number of augmented graphs generated per training graph. These augmented graphs are obtained using a graph augmentation generator $A_\lambda$, parameterized by $\lambda$ as a mapping $A_\lambda : \mathcal{G}_n, y_n \to A_\lambda(\mathcal{G}_n, y_n) \in \mathbb{G} \times \mathbb{Y}$, where $\mathbb{G}$ denotes the space of all possible graphs, and $\mathbb{Y}$ is the label space. The generator $A_\lambda$ may be either deterministic or stochastic with a dependence on a prior distribution $\mathcal{P}(\lambda)$. Examples of such augmentation strategies can be found in Appendix G.

We use the notation $\widetilde{\mathcal{G}}_n^m \sim A_\lambda$ to represent an augmented graph sampled from the augmentation strategy $A_\lambda$, i.e., $(\widetilde{\mathcal{G}}_n^m, \widetilde{y}_n^m) \sim A_\lambda(\mathcal{G}_n, y_n)$. With the augmented data, the loss function is modified to account for multiple augmented versions of each graph,

$$\mathcal{L}^{\text{aug}} = \frac{1}{N} \sum_{n=1}^{N} \mathbb{E}_{\widetilde{\mathcal{G}}_n^m \sim A_\lambda} \left[ \ell(f(\widetilde{\mathcal{G}}_n^m, \theta), \widetilde{y}_n^m) \right].$$

For simplicity, we denote the loss for the original graph by $\ell(f(\mathcal{G}_n, \theta), y_n) = \ell(\mathcal{G}_n, \theta)$ and the loss for an augmented graph as,

$$\mathbb{E}_{\widetilde{\mathcal{G}}_n^m \sim A_\lambda} \left[ \ell(f(\widetilde{\mathcal{G}}_n^m, \theta), \widetilde{y}_n^m) \right] = \ell^{\text{aug}}(\widetilde{\mathcal{G}}_n, \theta).$$

Via the law of large numbers, $\mathcal{L}^{\text{aug}}$ is empirically estimated,

$$\mathcal{L}^{\text{aug}} = \frac{1}{N} \sum_{n=1}^{N} \ell^{\text{aug}}(\widetilde{\mathcal{G}}_n, \theta)$$

$$\simeq \frac{1}{NM} \sum_{n=1}^{N} \sum_{m=1}^{M} \ell(f(\widetilde{\mathcal{G}}_n^m, \theta), \widetilde{y}_n^m).$$

To understand the impact of data augmentation on the graph classification performance, we analyze the effect of the augmentation strategy on the generalization risk $\mathbb{E}_{\mathcal{G}\sim\mathcal{D}}[\ell(\mathcal{G},\theta)]$. More specifically, we want to study the generalization error,

$$\eta = \mathbb{E}_{\mathcal{G}\sim\mathcal{D}}[\ell(\mathcal{G},\theta_{\text{aug}})] - \mathbb{E}_{\mathcal{G}\sim\mathcal{D}}[\ell(\mathcal{G},\theta_{\star})],$$

where $\theta_{\text{aug}}$ and $\theta_{\star}$ are the optimal GNN parameters for the augmented and non-augmented settings, respectively,

$$\theta_{\star} = \arg\min_{\theta}\mathcal{L}_{\theta}, \quad \theta_{\text{aug}} = \arg\min_{\theta}\mathcal{L}_{\theta}^{\text{aug}},$$

which can be estimated empirically as follows,

$$\hat{\theta} = \arg\min_{\theta}\frac{1}{N}\sum_{n=1}^{N}\ell(\mathcal{G}_n,\theta),$$

$$\hat{\theta}_{\text{aug}} = \arg\min_{\theta}\frac{1}{NM}\sum_{n=1}^{N}\sum_{m=1}^{M}\ell(\widetilde{\mathcal{G}}_n^m,\theta).$$

By theoretically studying the generalization error $\eta$, we aim to quantify the effect of each augmentation strategy on the overall classification performance, providing insights into the benefits and potential trade-offs of data augmentation in graph-based learning tasks. In Theorem 3.1, we present a regret bound of the generalization error using Rademacher complexity defined as follows (Yin et al., 2019),

$$\mathcal{R}(\ell) = \mathbb{E}_{\epsilon_n\sim P_\epsilon}\left[\sup_{\theta\in\Theta}\left|\frac{1}{N}\sum_{n=1}^{N}\epsilon_n\ell(\mathcal{G}_n,\theta)\right|\right],$$

where $\epsilon_n$ are independent Rademacher variables, taking values $+1$ or $-1$ with equal probability, $P_\epsilon$ is the Rademacher distribution, and $\Theta$ is the hypothesis class. Rademacher complexity is a fundamental concept in statistical learning, which indicates how well a learned function will perform on unseen data (Shalev-Shwartz & Ben-David, 2014). Intuitively, the Rademacher complexity measures the capacity of a GNN to fit random noise (Zhu et al., 2009), where a lower Rademacher complexity indicates better generalization.

**Theorem 3.1.** *Let $\ell$ be a classification loss function with $L_{Lip}$ as a Lipschitz constant and $\ell(\cdot,\cdot)\in[0,1]$. Then, with a probability at least $1-\delta$ over the samples $\mathcal{D}_{train}$, we have,*

$$\mathbb{E}_{\mathcal{G}\sim\mathcal{D}}\left[\ell(\mathcal{G},\hat{\theta}_{aug})\right] - \mathbb{E}_{\mathcal{G}\sim\mathcal{D}}[\ell(\mathcal{G},\theta_{\star})] \leq 2\mathcal{R}(\ell_{aug}) +$$

$$5\sqrt{\frac{2\log(4/\delta)}{N}} + 2L_{Lip}\mathbb{E}_{\mathcal{G}\sim\mathcal{D},\widetilde{\mathcal{G}}\sim A_\lambda}\left[\left\|\widetilde{\mathcal{G}}-\mathcal{G}\right\|\right].$$

*Moreover, we have,*

$$\mathcal{R}(\ell_{aug}) \leq \mathcal{R}(\ell) + \max_{n}L_{Lip}\mathbb{E}_{\widetilde{\mathcal{G}}_n^m\sim A_\lambda}\left[\left\|\widetilde{\mathcal{G}}_n^m-\mathcal{G}_n\right\|\right].$$

Theorem 3.1 relies on the assumption that the loss function is Lipschitz continuous. This assumption is realistic, given that the input node features and graph structures in real-world datasets are typically bounded, i.e., node features are typically normalized or constrained within a fixed range, while graph structures, represented by adjacency matrices or their normalized forms, have bounded spectral properties, ensuring a constrained input space. Additionally, we can ensure that the loss function is bounded within $[0,1]$ by composing any standard classification loss with a strictly increasing function that maps values to the interval $[0,1]$. We provide the proof of this theorem in Appendix A.

A direct implication of Theorem 3.1 is that if we chose the right data augmentation strategy $A_\lambda$ that minimizes the expected distance between original graphs and augmented ones $\mathbb{E}_{\mathcal{G}\sim\mathcal{D},\widetilde{\mathcal{G}}\sim A_\lambda}\left[\left\|\widetilde{\mathcal{G}}-\mathcal{G}\right\|\right]$, we can guarantee with a high probability that the data augmentation decreases both the Rademacher complexity and the generalization risk. Specifically, we need enough diversity to ensure a non-zero difference in the Rademacher complexity compared to the non-augmented case while controlling the expected distance so that we have a chance to reduce both the Rademacher complexity and the overall generalization error $\eta$. On the other hand, if the distance is large, we cannot guarantee that data augmentation will outperform the normal training setting.

The findings of Theorem 3.1 hold for all norms defined on the graph input space. Specifically, let us consider the graph structure space $(\mathbb{A},\|\cdot\|_{\mathbb{A}})$ and the feature space $(\mathbb{X},\|\cdot\|_{\mathbb{X}})$, where $\|\cdot\|_{\mathbb{A}}$ and $\|\cdot\|_{\mathbb{X}}$ denote the norms applied to the graph structure and features, respectively. Assuming a maximum number of nodes per graph, which is a realistic assumption for real-world data, the product space $\mathbb{A}\times\mathbb{X}$ is a finite-dimensional real vector space, and all the norms are equivalent. Thus, the choice of norm does not affect the theorem, as long as the Lipschitz constant is adjusted accordingly. Additional details on graph distance metrics and the comparison between the Lipschitz constants of GCN and GIN are provided in Appendices H and C.

The variety of distance metrics on the original graph space offers different upper bounds, which leads to distinct criteria for data augmentation based on these metrics to control the upper bound in Theorem 3.1. Instead, we propose to shift the focus to the hidden representation space of graphs, where we aim to derive more consistent and meaningful augmentations. If $L_{Lip}$ is taken to be the Lipschitz constant of the post-readout function only, then Theorem 3.1 still applies for the norm on the graph-level embeddings produced by the readout function, i.e., $\|\mathbf{h}_{\widetilde{\mathcal{G}}}-\mathbf{h}_{\mathcal{G}}\|$.

Working at the level of hidden representations of graphs, rather than directly in the graph input space, offers additional advantages. Hidden representations capture both the structural information and node features of each graph, en-

abling augmentation that enhances the generalization of both aspects simultaneously. Moreover, node alignment is needed to compare the original and augmented graphs, which is computationally expensive. By operating on hidden representations instead, node alignment becomes unnecessary. Furthermore, as we will discuss in Section 3.4, the effectiveness of augmented data depends on the specific GNN architecture. By leveraging graph representations learned through a GNN, we ensure that the augmentation process remains architecture-specific, aligning with the inductive biases of the chosen model.

### 3.2. Proposed Approach

Based on the theoretical findings, it is crucial to employ a data augmentation technique that effectively controls the term $\mathbb{E}_{\mathcal{G} \sim \mathcal{D}, \widetilde{\mathcal{G}} \sim A_\lambda} \left[ \|\mathbf{h}_{\widetilde{\mathcal{G}}} - \mathbf{h}_{\mathcal{G}}\| \right]$ which measures the expected deviation in the hidden representations of graphs under augmentation, to achieve stronger generalization guarantees. To better understand this term, we express it in terms of graph representations rather than graphs themselves,

$$\mathbb{E}_{\mathcal{G} \sim \mathcal{D}, \widetilde{\mathcal{G}} \sim A_\lambda} \left[ \|\mathbf{h}_{\widetilde{\mathcal{G}}} - \mathbf{h}_{\mathcal{G}}\| \right] = \mathbb{E}_{\mathbf{h} \sim \delta_{\mathcal{D}}, \widetilde{\mathbf{h}} \sim Q_\lambda} \left[ \|\widetilde{\mathbf{h}} - \mathbf{h}\| \right],$$

where $Q_\lambda$ represents the augmentation strategy at the level of hidden representations, replacing $A_\lambda$, which operates at the graph level, and $\delta_{\mathcal{D}} : \mathbf{h} \mapsto \frac{1}{N} \sum_{n=1}^{N} \delta_{\mathbf{h}_{\mathcal{G}_n}}(\mathbf{h})$ represents the Dirac distribution over the training graph representations, capturing the empirical distribution of graph embeddings.

Various universal approximators can be used as generators $Q_\lambda$, including generative models like Generative Adversarial Networks (GANs) (Yang et al., 2012). These models are capable of approximating any probability distribution, making them powerful tools for learning complex augmentations. However, we specifically choose Gaussian Mixture Models (GMMs), which are well-suited for this purpose, and can effectively approximate any data distribution, c.f. Theorem 3.2. GMMs are computationally fast compared to other generative approaches, making them suitable for large-scale graph datasets. As shown in Appendix E, GMM-based augmentation yields better results compared to alternative generative strategies. Moreover, due to the exponential decay of Gaussian distributions the expected distance $\mathbb{E}_{\mathbf{h} \sim \delta_{\mathcal{D}}, \widetilde{\mathbf{h}} \sim Q_\lambda} \left[ \|\mathbf{h} - \widetilde{\mathbf{h}}\| \right]$ is well-controlled.

**Theorem 3.2.** *(Goodfellow et al., 2016, Page 65) A Gaussian mixture model is a universal approximator of densities, in the sense that any smooth density can be approximated with any specific nonzero amount of error by a Gaussian mixture model with enough components.*

To achieve this, we first train a standard GNN on the graph classification task using the training set. Next, we obtain embeddings for all training graphs using the READOUT output, resulting in $\mathcal{H} = \{\mathbf{h}_{\mathcal{G}_n} \text{ s.t. } \mathcal{G}_n \in \mathcal{D}_{\text{train}}\}$.

---

**Algorithm 1** Graph classification with GRATIN

**Inputs:** GNN of $T$ layers $f(\cdot, \theta) = \Psi \circ \text{READOUT} \circ g$, where $g$ is the composition of message passing layers, i.e, $g = \cup_{t=0}^{T} \{\text{AGGREGATE}^{(t)} \circ \text{COMBINE}^{(t)}(\cdot)\}$ and $\Psi$ is the post-readout function, graph classification dataset $\mathcal{D}$, loss function $\mathcal{L}$;
**Steps:**
**1.** Train GNN $f$ on the training set $\mathcal{D}_{\text{train}}$;
**2.** Use the trained message passing layers and the readout function to generate graph representations $\mathcal{H} = \{\mathbf{h}_{\mathcal{G}_n} \ s.t. \ \mathcal{G}_n \in \mathcal{D}_{\text{train}}\}$ for the training set;
**3.** Partition the training set $\mathcal{D}_{\text{train}}$ by classes, such that $\mathcal{D}_{\text{train}} = \bigcup_c \mathcal{D}_c$ where $\mathcal{D}_c = \{\mathcal{G}_n \in \mathcal{D}_{\text{train}} , \ y_n = c\}$;
**foreach** $c \in \{0, \dots, C\}$ **do**
    **3.1.** Fit a GMM distribution $p_c$ on the graph representations $\mathcal{H}_c = \{\mathbf{h}_{\mathcal{G}_n} \text{ s.t. } \mathcal{G}_n \in \mathcal{D}_c\}$;
    **3.2.** Sample new graph representation $\widetilde{\mathcal{H}_c} = \{\widetilde{\mathbf{h}} \text{ s.t. } \widetilde{\mathbf{h}} \sim p_c\}$ from the distribution $p_c$;
    **3.3.** Include the sampled representations $\widetilde{\mathcal{H}_c}$ with trained representations $\mathcal{H}_c = \mathcal{H}_c \cup \widetilde{\mathcal{H}_c}$;
**end foreach**
**4.** Finetune the post-readout function $\Psi$ on the graph classification task directly on the new training set $\mathcal{H} = \cup_c \mathcal{H}_c$;

---

These embeddings are used as the basis for generating augmented training graphs. We then partition the training set $\mathcal{D}_{\text{train}}$ by classes, such that $\mathcal{D}_{\text{train}} = \bigcup_c \mathcal{D}_c$ where $\mathcal{D}_c = \{\mathcal{G}_n \in \mathcal{D}_{\text{train}} , \ y_n = c\}$. The objective is to learn new graph representations from these embeddings, and create augmented data for improved training.

We use the EM algorithm to learn the best-fitting GMM for the embeddings of each partition $\mathcal{D}_c$, denoted as $\mathcal{H}_c = \{\mathbf{h}_{\mathcal{G}_n} \text{ s.t. } \mathcal{G}_n \in \mathcal{D}_c\}$. The EM algorithm finds maximum likelihood estimates for each cluster $\mathcal{H}_c$, following the procedure described in (Bishop & Nasrabadi, 2006).

Once a GMM distribution $p_c$ is fitted for each partition $\mathcal{D}_c$, we use this GMM to generate new augmented data by sampling hidden representations from $p_c$. Each new sample drawn from $p_c$ is then assigned the corresponding partition label $c$, ensuring that the augmented data inherits the label structure from the original partitions. After merging the hidden representations of both the original training data and the augmented graph data, we fine-tune the post-readout function, i.e., the final part of the GNN, which occurs after the readout function, on the graph classification task. Since the post-readout function consists of a linear layer followed by a Softmax function, the finetuning process is relatively fast. To evaluate our model during inference on test graphs, we input the test graphs into the GNN layers trained in the initial step to compute the hidden graph representations. For the post-readout function, we use the weights obtained from the second stage of training. Algorithm 1 and Figure 1 provide a summary of the GRATIN model.

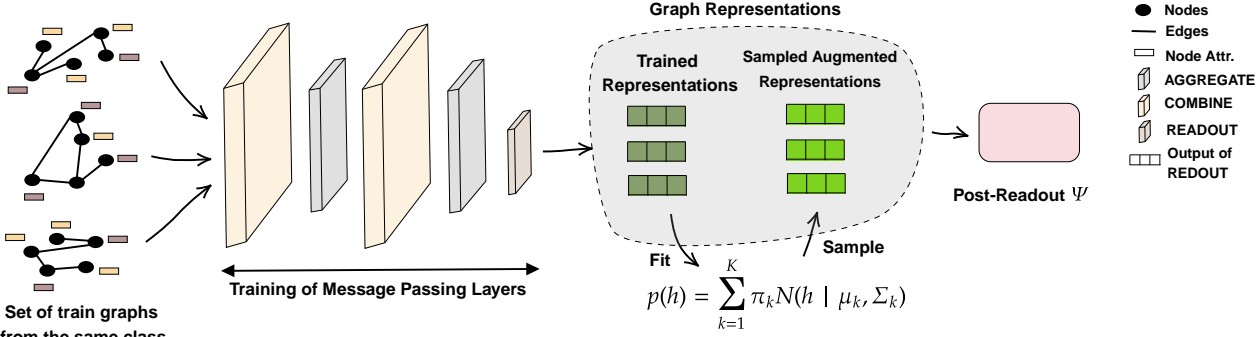

*Figure 1.* Illustration of GRATIN. *Step 1*. We first train the GNN on the graph classification task using the training graphs. *Step 2*. Next, we utilize the weights from the message passing layers to generate graph representations for the training graphs. *Step 3*. A GMM is then fit to these graph representations, from which we sample new graph representations. *Step 4*. Finally, we fine-tune the post-readout function for the graph classification task, using both the original training graphs and the augmented graph representations. For inference on the test set, we use the message passing weights trained in *Step 1* and the post-readout function weights trained in *Step 4*.

### 3.3. Time Complexity

One advantage of GRATIN is its efficiency, as it generates new augmented graph representations with minimal computational time. Unlike baseline methods, which apply augmentation strategies to each individual training graph (or pair of graphs in Mixup-based approaches) separately, our method learns the distribution of graph representations across the entire training dataset simultaneously using the EM algorithm (Ng, 2000). If $N = |\mathcal{D}_{\text{train}}|$ is the number of training graphs in the dataset, $d$ is the dimension of graph hidden representations $\{\mathbf{h}_\mathcal{G}, \mathcal{G} \in \mathcal{D}_{\text{train}}\}$, and $K$ is the number of Gaussian components in the GMM, then the complexity to fit a GMM on $T$ iterations is $\mathcal{O}(N \cdot K \cdot T \cdot d^2)$ (Yang et al., 2012). We compare the data augmentation times of our approach and the baselines in Table 7. Due to our different training scheme, i.e., where we first train the message passing layers and then train the post-readout function after learning the GMM distribution, we have measured the total backpropagation time and compared it with the backpropagation time of the baseline methods. The training time of baseline models varies depending on the augmentation strategy used, specifically whether it involves pairs of graphs or individual graphs. Even in cases where a graph augmentation has a low computational cost for some baselines, training can still be time-consuming as multiple augmented graphs are required to achieve satisfactory test accuracy. In contrast, GRATIN generates only one augmented graph per training graph, demonstrating effective generalization on the test set. Overall, our data augmentation approach is highly efficient during the sampling of augmented data, with minimal impact on the training time. A complete analysis of the time complexity of GRATIN and the baselines can be found in Appendix F.

### 3.4. Analyzing the Generalization Ability of the Augmented Graphs via Influence Functions

We use *influence functions* (Law, 1986; Koh & Liang, 2017; Kong et al., 2021) to understand the impact of augmented data on the model performance on the test set and thus motivate the use of a data augmentation strategy, which is specific to the model architecture and model weights.

In Theorem 3.3, we derive a closed-form formula for the impact of adding an augmented graph $\widetilde{\mathcal{G}}_n^m$ on the GNN's performance on a test graph $\mathcal{G}_k^{\text{test}}$, where the GNN is trained solely on the training set, without the augmented graph.

**Theorem 3.3.** *Given a test graph $\mathcal{G}_k$, let $\hat{\theta} = \arg\min_\theta \mathcal{L}$ be the GNN parameters that minimize the objective function in* (1). *The impact of upweighting the objective function $\mathcal{L}$ to $\mathcal{L}_{n,m}^{aug} = \mathcal{L} + \epsilon_{n,m}\ell(\widetilde{\mathcal{G}}_n^m, \theta)$, where $\widetilde{\mathcal{G}}_n^m$ is an augmented graph candidate of the training graph $\mathcal{G}_n$ and $\epsilon_{n,m}$ is a sufficiently small perturbation parameter, on the model performance on the test graph $\mathcal{G}_k^{test}$ is given by*

$$\frac{d\ell(\mathcal{G}_k^{test}, \hat{\theta}_{\epsilon_{n,m}})}{d\epsilon_{n,m}} = -\nabla_\theta\ell(\mathcal{G}_k^{test}, \hat{\theta})\mathbf{H}_{\hat{\theta}}^{-1}\nabla_\theta\ell(\widetilde{\mathcal{G}}_n^m, \hat{\theta}),$$

*where $\hat{\theta}_{\epsilon_{n,m}} = \arg\min_\theta \mathcal{L}_{n,m}^{aug}$ denotes the parameters that minimize the upweighted objective function $\mathcal{L}_{n,m}^{aug}$ and $\mathbf{H}_{\hat{\theta}} = \nabla_\theta^2\mathcal{L}(\hat{\theta})$ is the Hessian matrix of the loss w.r.t. the model parameters.*

We provide the proof of Theorem 3.3 in Appendix B. The influence scores are useful for evaluating the effectiveness of the augmented data on each test graph. The strength of influence function theory lies in its ability to analyze the effect of adding augmented data to the training set without actually retraining on this data. As noticed, these influence scores depend not only on the augmented graphs themselves

but also on the model's weights and architecture. This highlights the need for a graph data augmentation strategy tailored specifically to the GNN backbone in use, as opposed to traditional techniques like DropNode, DropEdge, and $\mathcal{G}$-Mixup, which are general-purpose methods that can be applied with any GNN architecture.

Theorem 3.3 is valid for any differentiable loss function. More specifically, if the chosen loss is the cross entropy or the negative log-likelihood, then the Hessian matrix corresponds to the Fisher information matrix (Barshan et al., 2020; Lee et al., 2022). Consequently, the norm of $\mathbf{H}_{\hat{\theta}}^{-1}$, i.e., the inverse of the Hessian matrix, can be bounded above using the Cramér–Rao inequality (Nielsen, 2013). Therefore, a trivial case where the norm of influence scores is zero arises when the gradient of the loss function with respect to the input graphs vanishes. This scenario, for instance, can occur in the DD dataset when using GIN. A detailed analysis of this phenomenon is provided in Section 4. In these cases, data augmentation becomes ineffective, having minimal impact on the GNN's ability to generalize. We can measure the average influence $\mathcal{I}(\widetilde{\mathcal{G}}_n^m)$ of an augmented graph $\widetilde{\mathcal{G}}_n^m$ on the test set by averaging the derivatives as follows,

$$\mathcal{I}(\widetilde{\mathcal{G}}_n^m) = \frac{-1}{|\mathcal{D}_{\text{test}}|} \sum_{\mathcal{G}_k^{\text{test}} \in \mathcal{D}_{\text{test}}} \frac{d\ell(\mathcal{G}_k^{\text{test}}, \hat{\theta}_{\epsilon_{n,m}})}{d\epsilon_{n,m}}.$$

A negative value of $\mathcal{I}(\widetilde{\mathcal{G}}_n^m)$ indicates that adding the augmented data to the training set would increase the prediction loss on the test set, negatively affecting the GNN's generalization. In contrast, a good augmented graph is one with a positive $\mathcal{I}(\widetilde{\mathcal{G}}_n^m)$, indicating improved generalization. In Figure 2, we present the density of the average influence scores of each augmented data on the test set.

### 3.5. Fisher-Guided GMM Augmentation

Using influence scores, we can further improve the generalization of the GNN by filtering candidate augmented representations. The process consists of three key stages. *(i) Primary GNN training:* The GNN model is first trained on the original training set without incorporating any augmented graphs. *(ii) Augmentation and filtering:* A pool of candidate augmented graph representations is generated using a data augmentation strategy based on GMMs. Because computing the gradient $\nabla_\theta \ell(\mathcal{G}_k^{\text{test}}, \hat{\theta})$ requires access to ground-truth labels, we evaluate the influence of each candidate augmented graph using the set of validation graph rather than on the unseen test set. This yields a ranking of augmented graphs by their estimated impact on validation performance. During this step, we compute both the gradient and the Hessian only with respect to the post-readout parameters. *(iii) Filtering:* Finally, we combine a subset of the highest-ranked augmented graphs with the original

*Table 1.* Classification accuracy ($\pm$ std) on different benchmark graph classification datasets for the data augmentation baselines based on the GCN backbone. The higher the accuracy (in %) the better the model. Highlighted are the **first**, second best results.

| Model | IMDB-BIN | IMDB-MUL | MUTAG | PROTEINS | DD |
|---|---|---|---|---|---|
| No Aug. | $73.00_{\pm4.94}$ | $47.73_{\pm2.64}$ | $73.92_{\pm5.09}$ | $69.99_{\pm5.35}$ | $69.69_{\pm2.89}$ |
| DropEdge | $71.70_{\pm5.42}$ | $45.67_{\pm2.46}$ | $73.39_{\pm8.86}$ | $70.07_{\pm3.86}$ | $69.35_{\pm3.37}$ |
| DropNode | $\mathbf{74.00_{\pm3.44}}$ | $43.80_{\pm3.54}$ | $73.89_{\pm8.53}$ | $69.81_{\pm4.61}$ | $69.01_{\pm3.95}$ |
| SubMix | $\underline{72.70_{\pm5.59}}$ | $46.00_{\pm2.44}$ | $\underline{77.13_{\pm9.69}}$ | $67.57_{\pm4.56}$ | $70.11_{\pm4.48}$ |
| $\mathcal{G}$-Mixup | $72.10_{\pm3.27}$ | $48.33_{\pm3.06}$ | $\mathbf{88.77_{\pm5.71}}$ | $65.68_{\pm5.03}$ | $61.20_{\pm3.88}$ |
| GeoMix | $69.69_{\pm3.37}$ | $\underline{49.80_{\pm4.71}}$ | $74.39_{\pm7.37}$ | $69.63_{\pm5.37}$ | $68.50_{\pm3.74}$ |
| GRATIN | $71.00_{\pm4.40}$ | $\mathbf{49.82_{\pm4.26}}$ | $76.05_{\pm6.74}$ | $\mathbf{70.97_{\pm5.07}}$ | $\mathbf{71.90_{\pm2.81}}$ |

*Table 2.* Classification accuracy ($\pm$ std) on different benchmark graph classification datasets for the data augmentation baselines based on the GIN backbone. The higher the accuracy (in %) the better the model. Highlighted are the **first**, second best results.

| Model | IMDB-BIN | IMDB-MUL | MUTAG | PROTEINS | DD |
|---|---|---|---|---|---|
| No Aug. | $70.30_{\pm3.66}$ | $\underline{48.53_{\pm4.05}}$ | $83.42_{\pm2.12}$ | $69.54_{\pm3.61}$ | $68.00_{\pm3.18}$ |
| DropEdge | $70.40_{\pm4.03}$ | $46.80_{\pm3.91}$ | $74.88_{\pm9.62}$ | $68.27_{\pm5.21}$ | $67.82_{\pm4.46}$ |
| DropNode | $70.30_{\pm3.49}$ | $45.20_{\pm4.24}$ | $75.53_{\pm7.89}$ | $65.40_{\pm4.71}$ | $\mathbf{69.01_{\pm3.95}}$ |
| SubMix | $\mathbf{72.50_{\pm4.98}}$ | $48.13_{\pm2.12}$ | $81.90_{\pm9.21}$ | $\underline{70.44_{\pm2.58}}$ | $68.59_{\pm5.04}$ |
| $\mathcal{G}$-Mixup | $70.70_{\pm3.10}$ | $47.73_{\pm4.95}$ | $\underline{87.77_{\pm7.48}}$ | $68.82_{\pm3.48}$ | $63.91_{\pm2.09}$ |
| GeoMix | $70.60_{\pm4.61}$ | $47.20_{\pm3.75}$ | $81.90_{\pm7.55}$ | $69.80_{\pm5.33}$ | $68.34_{\pm5.30}$ |
| GRATIN | $\underline{71.70_{\pm4.24}}$ | $\mathbf{49.20_{\pm2.06}}$ | $\mathbf{88.83_{\pm5.02}}$ | $\mathbf{71.33_{\pm5.04}}$ | $\underline{68.61_{\pm4.62}}$ |

training set to finetune the post-readout function. This filtering setup aligns perfectly with the assumptions of Theorem 3.3, as we first train the post-readout function without any augmentation, then evaluate each augmentation's influence, and only afterward retrain the post-readout layer using the selected augmented graphs. Our experiments in Section 4 demonstrate that this training paradigm improves generalization across various datasets and GNN architectures.

## 4. Experimental Results

In this section, we present our results and analysis. Our experimental setup is described in Appendix J.

**On the Generalization of GNNs.** In Tables 1 and 2, we compare the test accuracy of our data augmentation strategy against baseline methods. Additional results for the same experiment on larger datasets can be found in Appendix K. We trained all baseline models using the same train/validation/test splits, GNN architectures, and hyperparameters to ensure a fair comparison. It is worth noting that the baselines exhibit high standard deviations, which is a common characteristic in graph classification tasks. Unlike node classification, graph classification is known to have a larger variance in performance metrics (Errica et al., 2020; Duval & Malliaros, 2022). Overall, our proposed approach consistently achieves the best or highly competitive performance for most of the datasets.

Additionally, we observed that the results of the baseline methods vary depending on the GNN backbone, motivating further investigation using influence functions. As demon-

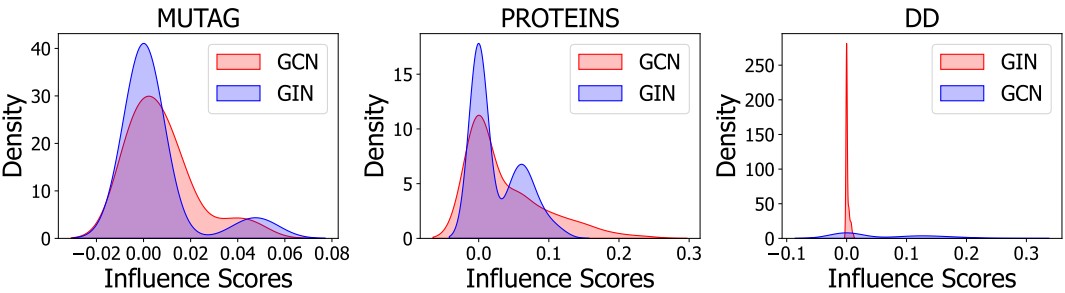

*Figure 2.* The density of the average influence scores of each augmented data on the test set.

*Table 3.* Robustness against structure corruption: We present the Classification accuracy ($\pm$ std). The higher the accuracy (in %) the better the model. We highlighted the best data augmentation strategy **bold**. For this experiment, we use the GCN backbone.

| Noise Budget | 10% | | | | 20% | | | |
|---|---|---|---|---|---|---|---|---|
| Dataset | IMDB-BIN | IMDB-MUL | PROTEINS | DD | IMDB-BIN | IMDB-MUL | PROTEINS | DD |
| DropNode | $66.40_{\pm5.51}$ | $44.46_{\pm2.13}$ | $69.18_{\pm4.87}$ | $65.79_{\pm3.23}$ | $64.80_{\pm5.01}$ | $43.06_{\pm2.86}$ | $67.73_{\pm6.43}$ | $64.35_{\pm4.56}$ |
| DropEdge | $66.70_{\pm5.10}$ | $43.80_{\pm3.11}$ | $69.36_{\pm5.90}$ | $68.42_{\pm4.76}$ | $63.20_{\pm6.30}$ | $41.80_{\pm3.15}$ | $68.10_{\pm5.05}$ | $67.06_{\pm2.53}$ |
| SubMix | $69.30_{\pm3.76}$ | $46.73_{\pm2.67}$ | $69.80_{\pm4.73}$ | $68.04_{\pm7.64}$ | $63.70_{\pm5.64}$ | $43.73_{\pm3.60}$ | $69.09_{\pm4.58}$ | $59.18_{\pm6.29}$ |
| GeoMix | $72.20_{\pm5.19}$ | $49.20_{\pm4.31}$ | $70.25_{\pm4.75}$ | $68.00_{\pm3.64}$ | $70.90_{\pm3.85}$ | $48.86_{\pm5.18}$ | $68.36_{\pm6.01}$ | $67.31_{\pm3.91}$ |
| $\mathcal{G}$-Mixup | $68.30_{\pm5.13}$ | $45.53_{\pm4.12}$ | $61.71_{\pm5.81}$ | $51.26_{\pm8.76}$ | $63.20_{\pm5.54}$ | $44.00_{\pm4.63}$ | $46.63_{\pm5.05}$ | $43.71_{\pm7.12}$ |
| NoisyGNN | $70.50_{\pm4.71}$ | $40.66_{\pm3.12}$ | $69.45_{\pm4.32}$ | $64.18_{\pm5.71}$ | $63.50_{\pm5.43}$ | $38.66_{\pm4.12}$ | $69.99_{\pm3.78}$ | $63.24_{\pm5.02}$ |
| GRATIN | $\mathbf{72.80_{\pm2.99}}$ | $\mathbf{49.36_{\pm4.53}}$ | $\mathbf{70.61_{\pm4.30}}$ | $\mathbf{68.68_{\pm3.72}}$ | $\mathbf{73.10_{\pm3.04}}$ | $\mathbf{49.53_{\pm3.54}}$ | $\mathbf{70.32_{\pm4.04}}$ | $\mathbf{69.01_{\pm3.09}}$ |

strated in Theorem 3.3, the gradient, and more generally, the model architecture, significantly influence how augmented data impacts the model's performance on the test set.

**Robustness to Structure Corruption.** Besides generalization, we assess the robustness of our data augmentation strategy, following the methodology outlined by (Zeng et al., 2024). Specifically, we test the robustness of data augmentation strategies against graph structure corruption by randomly removing or adding 10% or 20% of the edges in the training set. By corrupting only the training graphs, we introduce a distributional shift between the training and testing datasets. This approach allows us to evaluate GRATIN's ability to generalize well and predict the labels of test graphs, which can be considered OOD examples. The results of these experiments are presented in Table 3 for the IMDB-BIN, IMDB-MUL, PROTEINS, and DD datasets. As noted, our data augmentation strategy exhibits the best test accuracy in all cases and improves model robustness against structure corruption.

**Influence Functions.** In Figure 2, we show the density distribution of the average influence of augmented data sampled using GRATIN. These findings are consistent with the empirical results presented in Tables 1 and 2. For the MUTAG and PROTEINS datasets, we observe that GRATIN's data augmentation has a positive impact on both GCN and GIN models. In contrast, for the DD dataset, GRATIN shows no effect on GIN, while it generates many augmented samples with positive values of the influence

scores on GCN, thereby enhancing its performance. This behavior is consistent with the baselines, as most graph data augmentation strategies tend to enhance test accuracy more significantly for GCN than for GIN when applied to DD. This is an interesting phenomenon and worthy of deeper analysis. For the DD dataset, when using the GIN model at inference, we observe Softmax saturation, where the predicted class probabilities approach extreme values (close to 0 or 1), c.f. Appendix L. We suspect this saturation to happen due to the large average number of nodes in DD and the fact that GIN does not normalize the node representations, which may lead to a graph representation with large norms. This saturation results in the model making predictions with very high confidence. Consequently, the gradient of the loss function with respect to the input graphs eventually converges to 0. In such cases, the influence scores become negligible, as explained in Section 3.4.

**Fisher Based Filtering.** Figure 3 illustrates the impact of removing augmented representations on test accuracy in the Fisher-Guided GMM Augmentation experiment. The results presented are from a single training run. At the beginning, removing augmented graphs with low or negative influence scores improves generalization. The highest test accuracy is reached when a significant portion of low-quality augmentations has been removed while retaining high-influence ones. This indicates that a well-selected augmentation subset enhances model performance. As more augmentations are removed, the overall diversity of the training set decreases.

Since data augmentation generally helps the model generalize better, excessive removal reduces its effectiveness. At 100% removal, augmentation is entirely disabled, meaning the model is trained only on the original dataset, i.e., the reference case, leading to a significant drop in accuracy.

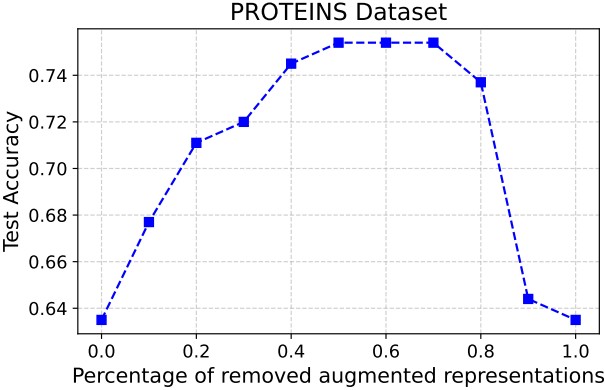

*Figure 3.* Effect of filtering augmented representations on test accuracy.

**Configuration Models.** As part of an ablation study, we propose a simple yet effective graph augmentation strategy inspired by *configuration models* (Newman, 2013). As shown in Theorem 3.1, the objective is to control the term $\mathbb{E}_{\mathcal{G}\sim\mathcal{D},\widetilde{\mathcal{G}}\sim A_\lambda}\left[\|\mathbf{h}_{\widetilde{\mathcal{G}}} - \mathbf{h}_\mathcal{G}\|\right]$, which can be achieved by regulating the distance between the original and the sampled graph within the input manifold, i.e., $\mathbb{E}_{\mathcal{G}\sim\mathcal{D},\widetilde{\mathcal{G}}\sim A_\lambda}\left[\|\widetilde{\mathcal{G}} - \mathcal{G}\|\right]$. The approach involves generating a sampled version of each training graph by randomly breaking existing edges into *half-edges* with probability $r$ and then randomly connecting half-edges until all edges are connected. The strength of this method lies in its simplicity and in preserving the degree distribution. If the distance norm is the $L_1$ distance between adjacency matrices, $|\mathcal{E}|r^2$ is an upper bound of $\mathbb{E}_{\mathcal{G}\sim\mathcal{D},\widetilde{\mathcal{G}}\sim A_\lambda}\left[\|\widetilde{\mathcal{G}} - \mathcal{G}\|\right]$, where $|\mathcal{E}|$ is the average number of edges. The results of this experiment are available in Appendix D.

## 5. Conclusion

We introduced GRATIN, a novel approach for graph data augmentation that enhances both the generalization and robustness of GNNs. Our method uses Gaussian Mixture Models (GMMs) applied at the output level of the Readout function, an approach motivated by theoretical findings. Using the universal approximation property of GMMs, we can sample new graph representations to effectively control the upper bound of the Rademacher complexity, ensuring improved generalization of GNNs. Through extensive experiments on widely used datasets, we demonstrated that our approach not only exhibits strong generalization ability but

also maintains robustness against structural perturbations. An additional advantage of our approach is its efficiency in terms of time complexity. Unlike baselines that generate augmented data for each individual or pair of training graphs, GRATIN fits the GMM to the entire training dataset at once, allowing for fast graph data augmentation without incurring significant additional backpropagation time.

## Acknowledgment

Y.A. and M.V. are supported by the French National Research Agency (ANR) via the AML-HELAS (ANR-19-CHIA-0020) project. Y.A. and J.L. are supported by the French National Research Agency (ANR) via the "GraspGNNs" JCJC grant (ANR-24-CE23-3888). F.M. acknowledges the support of the Innov4-ePiK project managed by the French National Research Agency under the 4th PIA, integrated into France2030 (ANR-23-RHUS-0002). This work was granted access to the HPC resources of IDRIS under the allocation "2024-AD010613410R2" made by GENCI.

## Impact Statement

This paper presents work whose goal is to advance the field of Machine Learning. There are many potential societal consequences of our work, none which we feel must be specifically highlighted here.

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

# A. Proof of Theorem 3.1

In this section, we provide a detailed proof of Theorem 3.1, aiming to derive a theoretical upper bound for both the generalization gap and the Rademacher complexity.

**Theorem 3.1** Let $\ell$ be a classification loss function with $\mathrm{L}_{\mathrm{Lip}}$ as a Lipschitz constant and $\ell(\cdot,\cdot) \in [0,1]$. Then, with a probability at least $1 - \delta$ over the samples $\mathcal{D}_{\mathrm{train}}$, we have,

$$\mathbb{E}_{\mathcal{G}\sim\mathcal{D}}\left[\ell(\mathcal{G},\hat{\theta}_{\mathrm{aug}})\right] - \mathbb{E}_{\mathcal{G}\sim\mathcal{D}}\left[\ell(\mathcal{G},\theta_\star)\right] \leq 2\mathcal{R}(\ell_{\mathrm{aug}}) + 5\sqrt{\frac{2\log(4/\delta)}{N}} + 2\mathrm{L}_{\mathrm{Lip}}\mathbb{E}_{\mathcal{G}\sim\mathcal{D},\widetilde{\mathcal{G}}\sim A_\lambda}\left[\left\|\widetilde{\mathcal{G}} - \mathcal{G}\right\|\right].$$

Moreover, we have,

$$\mathcal{R}(\ell_{\mathrm{aug}}) \leq \mathcal{R}(\ell) + \max_n \mathrm{L}_{\mathrm{Lip}}\mathbb{E}_{\widetilde{\mathcal{G}}_n^m\sim A_\lambda}\left[\left\|\widetilde{\mathcal{G}}_n^m - \mathcal{G}_n\right\|\right].$$

*Proof.* We will decompose $\mathbb{E}_{\mathcal{G}\sim\mathcal{D}}\left[\ell(\mathcal{G},\hat{\theta}_{\mathrm{aug}})\right] - \mathbb{E}_{\mathcal{G}\sim\mathcal{D}}\left[\ell(\mathcal{G},\theta_\star)\right]$ into a finite sum of 5 terms as follows,

$$\mathbb{E}_{\mathcal{G}\sim\mathcal{D}}\left[\ell(\mathcal{G},\hat{\theta}_{\mathrm{aug}})\right] - \mathbb{E}_{\mathcal{G}\sim\mathcal{D}}\left[\ell(\mathcal{G},\theta_\star)\right] = u_1 + u_2 + u_3 + u_4 + u_5$$

where,

$$u_1 = \mathbb{E}_{\mathcal{G}\sim\mathcal{D}}\left[\ell(\mathcal{G},\hat{\theta}_{\mathrm{aug}})\right] - \mathbb{E}_{\mathcal{G}\sim\mathcal{D}}\left[\mathbb{E}_{\widetilde{\mathcal{G}}\sim A_\lambda}\left[\ell(\widetilde{\mathcal{G}},\hat{\theta}_{\mathrm{aug}})\right]\right],$$

$$u_2 = \mathbb{E}_{\mathcal{G}\sim\mathcal{D}}\left[\mathbb{E}_{\widetilde{\mathcal{G}}\sim A_\lambda}\left[\ell(\widetilde{\mathcal{G}},\hat{\theta}_{\mathrm{aug}})\right]\right] - \frac{1}{N}\sum_{n=1}^N \mathbb{E}_{\widetilde{\mathcal{G}}_n^m\sim A_\lambda}\left[\ell(\widetilde{\mathcal{G}}_n^m,\hat{\theta}_{\mathrm{aug}})\right],$$

$$u_3 = \frac{1}{N}\sum_{n=1}^N \mathbb{E}_{\widetilde{\mathcal{G}}_n^m\sim A_\lambda}\left[\ell(\widetilde{\mathcal{G}}_n^m,\hat{\theta}_{\mathrm{aug}})\right] - \frac{1}{N}\sum_{n=1}^N \mathbb{E}_{\widetilde{\mathcal{G}}_n^m\sim A_\lambda}\left[\ell(\widetilde{\mathcal{G}}_n^m,\hat{\theta}_\star)\right],$$

$$u_4 = \frac{1}{N}\sum_{n=1}^N \mathbb{E}_{\widetilde{\mathcal{G}}_n^m\sim A_\lambda}\left[\ell(\widetilde{\mathcal{G}}_n^m,\hat{\theta}_\star)\right] - \mathbb{E}_{\mathcal{G}\sim\mathcal{D}}\left[\mathbb{E}_{\widetilde{\mathcal{G}}\sim A_\lambda}\left[\ell(\widetilde{\mathcal{G}},\theta_\star)\right]\right],$$

$$u_5 = \mathbb{E}_{\mathcal{G}\sim\mathcal{D}}\left[\mathbb{E}_{\widetilde{\mathcal{G}}\sim A_\lambda}\left[\ell(\widetilde{\mathcal{G}},\theta_\star)\right]\right] - \mathbb{E}_{\mathcal{G}\sim\mathcal{D}}\left[\ell(\mathcal{G},\theta_\star)\right].$$

We upperbound each of the terms in the sum. We get,

$$u_1 + u_5 = \mathbb{E}_{\mathcal{G}\sim\mathcal{D}}\left[\ell(\mathcal{G},\hat{\theta}_{\mathrm{aug}})\right] - \mathbb{E}_{\mathcal{G}\sim\mathcal{D}}\left[\mathbb{E}_{\widetilde{\mathcal{G}}\sim A_\lambda}\left[\ell(\widetilde{\mathcal{G}},\hat{\theta}_{\mathrm{aug}})\right]\right] + \mathbb{E}_{\mathcal{G}\sim\mathcal{D}}\left[\mathbb{E}_{\widetilde{\mathcal{G}}\sim A_\lambda}\left[\ell(\widetilde{\mathcal{G}},\theta_\star)\right]\right] - \mathbb{E}_{\mathcal{G}\sim\mathcal{D}}\left[\ell(\mathcal{G},\theta_\star)\right]$$

$$\leq \left|\mathbb{E}_{\mathcal{G}\sim\mathcal{D}}\left[\ell(\mathcal{G},\hat{\theta}_{\mathrm{aug}})\right] - \mathbb{E}_{\mathcal{G}\sim\mathcal{D}}\left[\mathbb{E}_{\widetilde{\mathcal{G}}\sim A_\lambda}\left[\ell(\widetilde{\mathcal{G}},\hat{\theta}_{\mathrm{aug}})\right]\right]\right| + \left|\mathbb{E}_{\mathcal{G}\sim\mathcal{D}}\left[\mathbb{E}_{\widetilde{\mathcal{G}}\sim A_\lambda}\left[\ell(\widetilde{\mathcal{G}},\theta_\star)\right]\right] - \mathbb{E}_{\mathcal{G}\sim\mathcal{D}}\left[\ell(\mathcal{G},\theta_\star)\right]\right|$$

$$\leq 2\sup_{\theta\in\Theta}\left|\mathbb{E}_{\mathcal{G}\sim\mathcal{D}}\left[\mathbb{E}_{\widetilde{\mathcal{G}}\sim A_\lambda}\left[\ell(\widetilde{\mathcal{G}},\theta)\right]\right] - \mathbb{E}_{\mathcal{G}\sim\mathcal{D}}\left[\ell(\mathcal{G},\theta)\right]\right|$$

$$\leq 2\sup_{\theta\in\Theta}\left|\mathbb{E}_{\mathcal{G}\sim\mathcal{D}}\left[\mathbb{E}_{\widetilde{\mathcal{G}}\sim A_\lambda}\left[\ell(\widetilde{\mathcal{G}},\theta)\right] - \ell(\mathcal{G},\theta)\right]\right|$$

$$\leq 2\sup_{\theta\in\Theta}\left|\mathbb{E}_{\mathcal{G}\sim\mathcal{D}}\left[\mathbb{E}_{\widetilde{\mathcal{G}}\sim A_\lambda}\left[\ell(\widetilde{\mathcal{G}},\theta) - \ell(\mathcal{G},\theta)\right]\right]\right|$$

$$\leq 2\mathrm{L}_{\mathrm{Lip}}\sup_{\theta\in\Theta}\mathbb{E}_{\mathcal{G}\sim\mathcal{D}}\mathbb{E}_{\widetilde{\mathcal{G}}\sim A_\lambda}\left[\left\|\widetilde{\mathcal{G}} - \mathcal{G}\right\|\right].$$

For the term $u_4$, we apply McDiarmid's inequality; we consider the two sets $\{(\mathcal{G}_n,y_n)\}_{n=1}^N$ and $\{(\mathcal{G}_n',y_n')\}_{n=1}^N$ are identical except at the $k$-th element, i.e. for a fixed $k \in \{1,\ldots,N\}$ we have $\mathcal{G}_k' \neq \mathcal{G}_k$ and $\forall n \in \{1,\ldots,N\}$, $n \neq k \Rightarrow \mathcal{G}_n' = \mathcal{G}_n$. This setup allows us to bound the change in the expected loss when a single graph is replaced. Since the classification loss satisfy $\ell(\cdot) \in [0,1]$, we get,

$$\left| \frac{1}{N} \sum_{n=1}^{N} \mathbb{E}_{\widetilde{\mathcal{G}} \sim A_\lambda} \left[ \ell(\mathcal{G}_n, \theta) \right] - \frac{1}{N} \sum_{n=1}^{N} \mathbb{E}_{\widetilde{\mathcal{G}} \sim A_\lambda} \left[ \ell(\mathcal{G}'_n, \theta) \right] \right| = \frac{1}{N} \left| \sum_{n=1}^{N} \mathbb{E}_{\widetilde{\mathcal{G}} \sim A_\lambda} \left[ \ell(\mathcal{G}_n, \theta) - \ell(\mathcal{G}'_n, \theta) \right] \right|$$

$$\leq \frac{1}{N} \left| \mathbb{E}_{\widetilde{\mathcal{G}} \sim A_\lambda} \left[ \ell(\mathcal{G}_k, \theta) - \ell(\mathcal{G}'_k, \theta) \right] \right|$$

$$\leq 2/N.$$

The first equality is obtained by the fact that $\forall n \neq k, \quad \mathcal{G}_n = \mathcal{G}'_n$ and $\mathcal{G}_k \neq \mathcal{G}'_k$, the last inequality is obtained by the fact that $\ell(\cdot) \in [0, 1]$.

Thus,

$$\forall t > 0, \quad \mathbb{P}(u_4 \geq t) = \mathbb{P}\left( \frac{1}{N} \sum_{n=1}^{N} \mathbb{E}_{\widetilde{\mathcal{G}} \sim A_\lambda} \left[ \ell(\widetilde{\mathcal{G}}, \theta_\star) \right] - \mathbb{E}_{\mathcal{G} \sim \mathcal{D}} \left[ \mathbb{E}_{\widetilde{\mathcal{G}} \sim A_\lambda} \left[ \ell(\widetilde{\mathcal{G}}, \theta_\star) \right] \right] \geq t \right)$$

$$\leq \exp\left( -\frac{2t^2}{\sum_{n=1}^{N} 4/N^2} \right)$$

$$= \exp\left( -\frac{Nt^2}{2} \right).$$

Therefore, for $\delta \in ]0, 1]$, and for $t = \sqrt{2\log(1/\delta)/N}$, i.e., $exp\left( -\frac{Nt^2}{2} \right) = \delta$, we have,

$$\mathbb{P}\left( u_4 \geq \sqrt{2\log(1/\delta)/N} \right) \leq \delta.$$

Therefore,

$$\mathbb{P}\left( u_4 < \sqrt{\frac{2\log(1/\delta)}{N}} \right) = 1 - \mathbb{P}\left( u_4 \geq \sqrt{\frac{2\log(1/\delta)}{N}} \right) \geq 1 - \delta.$$

Thus, with a probability of at least $1 - \delta$,

$$u_4 \leq \sqrt{\frac{2\log(1/\delta)}{N}} < \sqrt{\frac{2\log(4/\delta)}{N}}.$$

Moreover, Rademacher complexity holds for $u_2$,

$$u_2 = \mathbb{E}_{\mathcal{G} \sim \mathcal{D}} \left[ \mathbb{E}_{\widetilde{\mathcal{G}} \sim A_\lambda} \left[ \ell(\widetilde{\mathcal{G}}, \hat{\theta}_{\mathrm{aug}}) \right] \right] - \frac{1}{N} \sum_{n=1}^{N} \mathbb{E}_{\widetilde{\mathcal{G}}_n^m \sim A_\lambda} \left[ \ell(\widetilde{\mathcal{G}}_n^m, \hat{\theta}_{\mathrm{aug}}) \right] \leq 2\mathcal{R}(\ell_{\mathrm{aug}}) + 4\sqrt{\frac{2\log(4/\delta)}{N}}.$$

The above inequality tells us that the true risk $\mathbb{E}_{\mathcal{G} \sim \mathcal{D}} \left[ \mathbb{E}_{\widetilde{\mathcal{G}} \sim A_\lambda} \left[ \ell(\widetilde{\mathcal{G}}, \hat{\theta}_{\mathrm{aug}}) \right] \right]$ is bounded by the empirical risk $\frac{1}{N} \sum_{n=1}^{N} \mathbb{E}_{\widetilde{\mathcal{G}}_n^m \sim A_\lambda} \left[ \ell(\widetilde{\mathcal{G}}_n^m, \hat{\theta}_{\mathrm{aug}}) \right]$ plus a term depending on the Rademacher complexity of the augmented hypothesis class and an additional term that decreases with the size of the sample $N$.

Additionally, since $\hat{\theta}_{\mathrm{aug}}$ is the optimal parameter for the loss $\frac{1}{N} \sum_{n=1}^{N} \mathbb{E}_{\widetilde{\mathcal{G}}_n^m \sim A_\lambda} \left[ \ell(\widetilde{\mathcal{G}}_n^m, \theta) \right]$, thus,

$$u_3 \leq 0.$$

By summing all the inequalities, we conclude that,

$$\mathbb{E}_{\mathcal{G} \sim \mathcal{D}} \left[ \ell(\mathcal{G}, \hat{\theta}_{\mathrm{aug}}) \right] - \mathbb{E}_{\mathcal{G} \sim \mathcal{D}} \left[ \ell(\mathcal{G}, \theta_\star) \right] < 2\mathcal{R}(\ell_{\mathrm{aug}}) + 5\sqrt{\frac{2\log(4/\delta)}{N}} + 2L_{\mathrm{Lip}} \mathbb{E}_{\mathcal{G} \sim \mathcal{D}} \mathbb{E}_{\widetilde{\mathcal{G}}_n^m \sim A_\lambda} \left[ \left\| \widetilde{\mathcal{G}}_n^m - \mathcal{G}_n \right\| \right].$$

Part 2 of the proof.

$$\mathcal{R}(\ell_{\text{aug}}) - \mathcal{R}(\ell) = \mathbb{E}_{\epsilon_n \sim P_\epsilon} \left[ \sup_{\theta \in \Theta} \left| \frac{1}{N} \sum_{n=1}^N \epsilon_n \ell_{\text{aug}}(\mathcal{G}_n, \theta) \right| - \sup_{\theta \in \Theta} \left| \frac{1}{N} \sum_{n=1}^N \epsilon_n \ell(\mathcal{G}_n, \theta) \right| \right]$$

$$\leq \mathbb{E}_{\epsilon_n \sim P_\epsilon} \left[ \sup_{\theta \in \Theta} \left| \frac{1}{N} \sum_{n=1}^N \epsilon_n \ell_{\text{aug}}(\mathcal{G}_n, \theta) - \frac{1}{N} \sum_{n=1}^N \epsilon_n \ell(\mathcal{G}_n, \theta) \right| \right]$$

$$= \mathbb{E}_{\epsilon_n \sim P_\epsilon} \left[ \sup_{\theta \in \Theta} \left| \frac{1}{N} \sum_{n=1}^N \epsilon_n \left( \ell_{\text{aug}}(\mathcal{G}_n, \theta) - \ell(\mathcal{G}_n, \theta) \right) \right| \right]$$

$$\leq \mathbb{E}_{\epsilon_n \sim P_\epsilon} \left[ \sup_{\theta \in \Theta} \frac{1}{N} \sum_{n=1}^N \left| \epsilon_n \left( \ell_{\text{aug}}(\mathcal{G}_n, \theta) - \ell(\mathcal{G}_n, \theta) \right) \right| \right]$$

$$\leq \sup_{\theta \in \Theta} \frac{1}{N} \sum_{n=1}^N \left| \ell_{\text{aug}}(\mathcal{G}_n, \theta) - \ell(\mathcal{G}_n, \theta) \right|$$

$$= \sup_{\theta \in \Theta} \frac{1}{N} \sum_{n=1}^N \left| \mathbb{E}_{\mathcal{G}_n^\lambda \sim A_\lambda} \left[ \ell(\mathcal{G}_n^\lambda, \theta) - \ell(\mathcal{G}_n, \theta) \right] \right|$$

$$\leq \max_{n \in \{1, \dots, N\}} \mathrm{L}_{\text{Lip}} \mathbb{E}_{\mathcal{G}_n^\lambda \sim A_\lambda} \left[ \left\| \mathcal{G}_n^\lambda - \mathcal{G}_n \right\| \right].$$

$\square$

## B. Proof of Theorem 3.3

In this section, we present the detailed proof of Theorem 3.3, which allows us to perform an in-depth theoretical analysis of our augmentation strategy through the lens of influence functions.

**Theorem 3.3** Given a test graph $\mathcal{G}_k$ from the test set, let $\hat{\theta} = \arg\min_\theta \mathcal{L}$ be the GNN parameters that minimize the objective function in (1). The impact of upweighting the objective function $\mathcal{L}$ to $\mathcal{L}_{n,m}^{\text{aug}} = \mathcal{L} + \epsilon_{n,m} \ell(\widetilde{\mathcal{G}}_n^m, \theta)$, where $\widetilde{\mathcal{G}}_n^m$ is an augmented graph candidate of the training graph $\mathcal{G}_n$ and $\epsilon_{n,m}$ is a sufficiently small perturbation parameter, on the model performance on the test graph $\mathcal{G}_k^{test}$ is given by

$$\frac{d\ell(\mathcal{G}_k^{\text{test}}, \hat{\theta}_{\epsilon_{n,m}})}{d\epsilon_{n,m}} = -\nabla_\theta \ell(\mathcal{G}_k^{test}, \hat{\theta}) \mathbf{H}_{\hat{\theta}}^{-1} \nabla_\theta \ell(\widetilde{\mathcal{G}}_n^m, \hat{\theta}),$$

where $\hat{\theta}_{\epsilon_{n,m}} = \arg\min_\theta \mathcal{L}_{n,m}^{\text{aug}}$ denotes the parameters that minimize the upweighted objective function $\mathcal{L}_{n,m}^{\text{aug}}$ and $\mathbf{H}_{\hat{\theta}} = \nabla_\theta^2 \mathcal{L}(\hat{\theta})$ is the Hessian Matrix of the loss w.r.t. the model parameters.

*Proof.* Let $\widetilde{\mathcal{G}}_n^m$ be an augmented graph candidate of the training graph $\mathcal{G}_n$ and $\epsilon_{n,m}$ is a sufficiently small perturbation parameter. The parameters $\hat{\theta}$ and $\hat{\theta}_{\epsilon_{n,m}}$ the parameters that minimize the empirical risk on the train set, i.e.,

$$\hat{\theta} = \arg\min_\theta \mathcal{L},$$

$$\hat{\theta}_{\epsilon_{n,m}} = \arg\min_\theta \mathcal{L}_{n,m}^{\text{aug}} = \arg\min_\theta \mathcal{L} + \epsilon_{n,m} \ell(\widetilde{\mathcal{G}}_n^m, \theta).$$

Therefore, we examine its first-order optimality conditions,

$$0 = \nabla_{\hat{\theta}} \mathcal{L} \tag{2}$$

$$0 = \nabla_{\hat{\theta}_{\epsilon_{n,m}}} \left( \mathcal{L} + \epsilon_{n,m} \ell(\widetilde{\mathcal{G}}_n^m, \theta) \right). \tag{3}$$

Using Taylor Expansion, we now develop Eq. (3). We have $\lim_{\epsilon_{n,m} \to 0} \hat{\theta}_{\epsilon_{n,m}} = \hat{\theta}$, thus,

$$0 \simeq \left[ \nabla_{\hat{\theta}} \mathcal{L}(\hat{\theta}) + \epsilon_{n,m} \nabla_{\hat{\theta}} \ell(\widetilde{\mathcal{G}}_n^m, \hat{\theta}) \right] + \left[ \nabla_{\hat{\theta}}^2 \mathcal{L}(\hat{\theta}) + \epsilon_{n,m} \nabla_{\hat{\theta}}^2 \ell(\widetilde{\mathcal{G}}_n^m, \hat{\theta}) \right] \left( \hat{\theta}_{\epsilon_{n,m}} - \hat{\theta} \right).$$

Therefore,

$$\hat{\theta}_{\epsilon_{n,m}} - \hat{\theta} = - \left[ \nabla_{\hat{\theta}}^2 \mathcal{L}(\hat{\theta}) + \epsilon_{n,m} \nabla_{\hat{\theta}}^2 \ell(\widetilde{\mathcal{G}}_n^m, \hat{\theta}) \right]^{-1} \left[ \nabla_{\hat{\theta}} \mathcal{L}(\hat{\theta}) + \epsilon_{n,m} \nabla_{\hat{\theta}} \ell(\widetilde{\mathcal{G}}_n^m, \hat{\theta}) \right].$$

Dropping the $\circ(\epsilon_{n,m})$ terms, and using the Equation 2, i.e. $\nabla_{\hat{\theta}} \mathcal{L} = 0$, we conclude that,

$$\frac{\hat{\theta}_{\epsilon_{n,m}} - \hat{\theta}}{\epsilon_{n,m}} = - \left[ \nabla_{\hat{\theta}}^2 \mathcal{L}(\hat{\theta}) \right]^{-1} \nabla_{\hat{\theta}} \ell(\widetilde{\mathcal{G}}_n^m, \hat{\theta}).$$

Therefore,

$$\frac{d\hat{\theta}_{\epsilon_{n,m}}}{d\epsilon_{n,m}} \simeq \frac{\hat{\theta}_{\epsilon_{n,m}} - \hat{\theta}}{\epsilon_{n,m}} = - \left[ \nabla_{\hat{\theta}}^2 \mathcal{L}(\hat{\theta}) \right]^{-1} \nabla_{\hat{\theta}} \ell(\widetilde{\mathcal{G}}_n^m, \hat{\theta}).$$

$$\frac{d\ell(\mathcal{G}_k^{test}, \hat{\theta}_{\epsilon_{n,m}})}{d\epsilon_{n,m}} = \frac{d\ell(\mathcal{G}_k^{test}, \hat{\theta}_{\epsilon_{n,m}})}{d\hat{\theta}_{\epsilon_{n,m}}} \frac{d\hat{\theta}_{\epsilon_{n,m}}}{d\epsilon_{n,m}}$$
$$= -\nabla_\theta \ell(\mathcal{G}_k^{test}, \hat{\theta}) \mathbf{H}_{\hat{\theta}}^{-1} \nabla_\theta \ell(\widetilde{\mathcal{G}}_n^m, \hat{\theta}).$$

$\square$

## C. Mathematical Expressions of GCN and GIN

In this section, we provide concise definitions of two widely used GNN architectures: Graph Convolutional Networks (GCN) and Graph Isomorphism Networks (GIN). These architectures differ in how they aggregate and combine information from neighboring nodes in a graph.

**Graph Convolutional Network (GCN) (Kipf & Welling, 2017).** The GCN updates node embeddings by aggregating normalized features from their neighbors. Specifically, for a node $v \in \mathcal{V}$, its feature vector $\mathbf{h}_v^{(t)}$ at layer $t$ is computed as,

$$\mathbf{h}_v^{(t)} = \sigma \left( \sum_{u \in \mathcal{N}(v) \cup \{v\}} \frac{1}{\sqrt{\deg(v) \deg(u)}} \mathbf{W}^{(t)} \mathbf{h}_u^{(t-1)} \right),$$

where $\mathbf{h}_u^{(t-1)}$ is the feature vector of node $u$ at layer $t - 1$, $\mathbf{W}^{(t)}$ is a trainable weight matrix for layer $t$, and $\sigma(\cdot)$ is a non-linear activation function, such as ReLU.

**Graph Isomorphism Network (GIN) (Xu et al., 2019).** GIN is designed to match the expressiveness of the Weisfeiler-Lehman (WL) graph isomorphism test. It updates a node's embedding by aggregating its own feature with those of its neighbors, followed by a a multi-layer perceptron (MLP). The update rule at each message passing layer $t$ is

$$\mathbf{h}_v^{(t)} = \text{MLP}^{(t)} \left( (1 + \epsilon^{(t)}) \cdot \mathbf{h}_v^{(t-1)} + \sum_{u \in \mathcal{N}(v)} \mathbf{h}_u^{(t-1)} \right),$$

where $\epsilon^{(t)}$ is a fixed or learnable scalar. GIN allows for more expressive feature transformations compared to methods with fixed aggregation schemes, enabling it to distinguish a broader range of graph structures.

**Matrix forms and comparison.** The node representations $\mathbf{h}_v^{(t)}$ at each layer can be expressed in matrix form as $\mathbf{H}^{(t)} \in \mathbb{R}^{p \times d_t}$, where $p$ is the number of nodes in the graph, $d_t$ is hidden dimension in the $t-$th layer, and $\mathbf{H}^{(t)}$ is the concatenation of all node representations $\mathbf{h}_v^{(t)}$ for $v \in \mathcal{V}$. For GCN, the update rule can be written as:

$$\mathbf{H}^{(t)} = \sigma\Big(\widetilde{\mathbf{D}}^{-1/2}\widetilde{\mathbf{A}}\widetilde{\mathbf{D}}^{-1/2}\mathbf{H}^{(t-1)}\mathbf{W}^{(t)}\Big),$$

where $\widetilde{\mathbf{A}} = \mathbf{A} + \mathbf{I}$ is the adjacency matrix with self-loops, and $\widetilde{\mathbf{D}}$ is its diagonal degree matrix. In contrast, for GIN, the update rule is given by:

$$\mathbf{H}^{(t)} = \mathrm{MLP}^{(t)}\Big((1 + \epsilon^{(t)})\mathbf{H}^{(t-1)} + \mathbf{A}\mathbf{H}^{(t-1)}\Big).$$

If the same MLP is used, comprising a learnable linear layer followed by a ReLU activation, the only difference between GCN and GIN lies in the *graph shift operator*: GCN uses the degree-normalized operator $\widetilde{\mathbf{D}}^{-1/2}\widetilde{\mathbf{A}}\widetilde{\mathbf{D}}^{-1/2}$, while GIN uses $(1 + \epsilon^{(t)})\mathbf{I} + \mathbf{A}$.

Therefore, when node features are taken as constant and only the graph structure is considered, the difference in their Lipschitz behavior can be traced back to the function that maps the adjacency matrix $A$ (and degrees) to these respective shift operators. In other words, the Lipschitz constant difference arises from whether the adjacency information is normalized $(\widetilde{\mathbf{D}}^{-1/2}\widetilde{\mathbf{A}}\widetilde{\mathbf{D}}^{-1/2})$ or as $((1 + \epsilon^{(t)})\mathbf{I} + \mathbf{A})$.

**Lipschitz constants of GCN and GIN.** Under constant node features, the Lipschitz constant of each architecture can be decomposed into two parts: one accounting for the sensitivity of the mapping from the adjacency matrix to the respective *graph shift operator*, i.e., degree-normalized $\widetilde{\mathbf{D}}^{-1/2}\widetilde{\mathbf{A}}\widetilde{\mathbf{D}}^{-1/2}$ for GCN vs. $(1 + \epsilon)\mathbf{I} + \mathbf{A}$ for GIN, and the other capturing all shared learnable transformations.

Let $\ell_{\mathrm{GCN}}$ and $\ell_{\mathrm{GIN}}$ be respectively the graph shift operators $A \mapsto \widetilde{\mathbf{D}}^{-1/2}\widetilde{\mathbf{A}}\widetilde{\mathbf{D}}^{-1/2}$ and $A \mapsto (1 + \epsilon)\mathbf{I} + \mathbf{A}$, and let $\ell_{\mathrm{params}}$ represent the product of Lipschitz factors arising from the shared functions. Then,

$$L_{\mathrm{GCN}} = \ell_{\mathrm{GCN}} \times \ell_{\mathrm{params}},$$
$$L_{\mathrm{GIN}} = \ell_{\mathrm{GIN}} \times \ell_{\mathrm{params}}.$$

It becomes clear that the difference between GCN and GIN Lipschitz constants depends solely on whether the graph adjacency is normalized, since $\ell_{\mathrm{params}}$ is common to both. It's straightforward that $\ell_{\mathrm{GIN}} \leq 1$ since the corresponding graph shift operator is just a translation. Let us now derive an upper bound for $\ell_{\mathrm{GIN}}$. Let $\mathbf{A}_1, \mathbf{A}_2 \in \mathbb{R}^{p \times p}$ be adjacency matrices of graphs on $n$ nodes. For each adjacency matrix $A_i$, we define the diagonal degree matrix, $\mathbf{D}_i = \mathrm{diag}\big(\deg_1^{(i)}, \ldots, \deg_p^{(i)}\big)$ where $\forall j \leq p, \ \deg_j^{(i)} = \sum_{k=1}^n (\mathbf{A}_i)_{jk}$. We have,

$$D_1^{-\frac{1}{2}}\mathbf{A}_1\mathbf{D}_1^{-\frac{1}{2}} - \mathbf{D}_2^{-\frac{1}{2}}\mathbf{A}_2\mathbf{D}_2^{-\frac{1}{2}} = \mathbf{D}_1^{-\frac{1}{2}}\mathbf{A}_1\mathbf{D}_1^{-\frac{1}{2}} - \mathbf{D}_1^{-\frac{1}{2}}\mathbf{A}_2\mathbf{D}_1^{-\frac{1}{2}} + \mathbf{D}_1^{-\frac{1}{2}}\mathbf{A}_2\mathbf{D}_1^{-\frac{1}{2}} - \mathbf{D}_2^{-\frac{1}{2}}\mathbf{A}_2\mathbf{D}_2^{-\frac{1}{2}}$$
$$= \underbrace{\mathbf{D}_1^{-\frac{1}{2}}\left(\mathbf{A}_1 - \mathbf{A}_2\right)\mathbf{D}_1^{-\frac{1}{2}}}_{(i)} + \underbrace{\left[\mathbf{D}_1^{-\frac{1}{2}} - \mathbf{D}_2^{-\frac{1}{2}}\right]\mathbf{A}_2\mathbf{D}_1^{-\frac{1}{2}} + \mathbf{D}_2^{-\frac{1}{2}}\mathbf{A}_2\left[\mathbf{D}_1^{-\frac{1}{2}} - \mathbf{D}_2^{-\frac{1}{2}}\right]}_{(ii)}.$$

We can upperbound each of (i), and (ii). For (i), we have the following upperbound,

$$\left\|\mathbf{D}_1^{-\frac{1}{2}}\left(\mathbf{A}_1 - \mathbf{A}_2\right)\mathbf{D}_1^{-\frac{1}{2}}\right\| \leq \|\mathbf{D}_1^{-\frac{1}{2}}\|^2 \|\mathbf{A}_1 - \mathbf{A}_2\| \leq \frac{1}{\delta_{1,\min}}\|\mathbf{A}_1 - \mathbf{A}_2\|,$$

where $\delta_{i,\min} = \min_j \deg_j^{(i)}$. For the second (ii), if we consider for example $\|\|$ as the $L_1$ norm, then,

$$\|(ii)\| \leq \|\mathbf{D}_1^{-1/2} - \mathbf{D}_2^{-1/2}\|\|\mathbf{A}_2\|\big(\|\mathbf{D}_1^{-1/2}\| + \|\mathbf{D}_2^{-1/2}\|\big),$$

$$\leq \frac{2}{\min(\delta_{1,\min}, \delta_{2,\min})^{1/2}}\|\mathbf{D}_1^{-1/2} - \mathbf{D}_2^{-1/2}\|\|\mathbf{A}_2\|$$

$$\leq \frac{2M}{\min(\delta_{1,\min}, \delta_{2,\min})^{1/2}}\|\mathbf{D}_1^{-1/2} - \mathbf{D}_2^{-1/2}\|$$

$$\leq \frac{2M}{\min(\delta_{1,\min}, \delta_{2,\min})^{5/2}}\|\mathbf{D}_1 - \mathbf{D}_2\|$$

$$\leq \frac{2M}{\min(\delta_{1,\min}, \delta_{2,\min})^{5/2}}\|\mathbf{A}_1\mathbf{1}_p - \mathbf{A}_2\mathbf{1}_p\|$$

$$\leq \frac{2Mp}{\min(\delta_{1,\min}, \delta_{2,\min})^{5/2}}\|\mathbf{A}_1 - \mathbf{A}_2\|,$$

where $M$ is the maximum norm of the adjacency matrix in the graph dataset, and $\mathbf{1}_p \in \mathbb{R}^p$ is the vector of ones.

Putting both inequalities together, we can come up with an upperbound for $\ell_{\text{GCN}}$ that depends on the minimum degree in the dataset.

## D. Configuration Models

In this section, we present a novel adaptation of configuration models as a graph data augmentation technique for GNN. Configuration models (Newman, 2013) enable the generation of randomized graphs that maintain the original degree distribution. We can, therefore, leverage this strategy to improve the generalization of GNNs. Below, we present the steps involved in our approach to using Configuration Models for Graph Data Augmentation:

1. **Extract edges:** For each training graph $\mathcal{G}_n$, we first extract the complete set of edges $\mathcal{E}_n$.

2. **Stub creation:** Using a Bernoulli distribution with parameter $r \in [0, 1]$, we randomly select a subset of candidate edges and *break* them to create *stubs* (half-edges).

3. **Stub pairing:** We then randomly pair these stubs to form new edges, creating a randomized graph structure with the same degree distribution.

Table 4 shows the performance of this approach on the two GNN backbones, GCN and GIN.

*Table 4.* Classification accuracy ($\pm$ std) on different benchmark graph classification datasets for the data augmentation baselines based on the GIN backbone. The higher the accuracy (in %) the better the model.

| Model | IMDB-BIN | IMDB-MUL | MUTAG | PROTEINS | DD |
|---|---|---|---|---|---|
| Config Models w/ GCN | $71.70_{\pm 3.16}$ | $48.40_{\pm 3.88}$ | $74.97_{\pm 6.77}$ | $70.08_{\pm 4.93}$ | $69.01_{\pm 3.44}$ |
| Config Models w/ GIN | $71.70_{\pm 4.24}$ | $49.00_{\pm 3.44}$ | $81.43_{\pm 10.05}$ | $68.34_{\pm 5.30}$ | $71.61_{\pm 5.93}$ |

As noticed, the configuration model-based graph augmentation method performs competitively with the baselines and even outperforms them in certain cases. This underscores the importance of Theorem 3.1. When compared to our approach GRATIN, the latter gives better results across different datasets and GNN backbones. This difference is primarily due to the configuration model based approach being model-agnostic, whereas GRATIN leverages the model's weights and architecture, as explained in Section 3.4 and supported by Theorem 3.3.

## E. Ablation Study

To provide additional comparison and motivate the use of GMMs with the EM algorithm within GRATIN, we expanded our evaluation to include additional methods for modeling the distribution of the graph representations. Specifically, the comparison includes:

- **GMM w/ Variational Bayesian Inference (VBI):** We specifically compared the Expectation-Maximization (EM) algorithm, discussed in the main paper, with the Variational Bayesian (VB) estimation technique for parameter estimation of each Gaussian Mixture Model (GMM) (Tzikas et al., 2008) for both the GCN and GIN models. The objective of including this baseline is to explore alternative approaches for fitting GMMs to the graph representations.

- **Kernel Density Estimation (KDE):** KDE is a Neighbor-Based Method and a non-parametric approach to estimating the probability density (Härdle et al., 2004). KDE estimates the probability density function by placing a kernel function (e.g., Gaussian) at each data point. The sum of these kernels approximates the underlying distribution. Sampling can be done using techniques like Metropolis-Hastings. The purpose of using KDE as a baseline is to evaluate alternative distributions different from the Gaussian Mixture Model (GMM).

- **Copula-Based Methods:** We model the dependence structure between variables using copulas, while marginal distributions are modeled separately. We sample from marginal distributions and then transform them using the copula (Nelsen, 2006).

- **Generative Adversarial Network (GAN):** GANs are powerful generative models that learn to approximate the data distribution through an adversarial process between two neural networks. To evaluate the performance of deep learning-based generative approaches for modeling graph representations, we included tGAN, a GAN architecture specifically designed for tabular data (Yang et al., 2012). We particularly train tGAN on the graph representations and then sample new graph representations from the generator.

*Table 5.* Ablation study on the density estimation scheme for learned GCN representations in GRATIN.

| Model | IMDB-BIN | IMDB-MUL | MUTAG | PROTEINS | DD |
|---|---|---|---|---|---|
| GMM w/ EM | $71.00_{\pm 4.40}$ | $49.82_{\pm 4.26}$ | $76.05_{\pm 6.47}$ | $70.97_{\pm 5.07}$ | $71.90_{\pm 2.81}$ |
| GMM w/ VBI | $71.00_{\pm 4.21}$ | $49.53_{\pm 4.26}$ | $76.05_{\pm 6.47}$ | $70.97_{\pm 4.52}$ | $71.64_{\pm 2.90}$ |
| KDE | $55.90_{\pm 10.29}$ | $39.53_{\pm 2.87}$ | $66.64_{\pm 6.79}$ | $59.56_{\pm 2.62}$ | $58.66_{\pm 3.97}$ |
| Copula | $69.80_{\pm 4.04}$ | $47.13_{\pm 3.45}$ | $74.44_{\pm 6.26}$ | $65.04_{\pm 3.37}$ | $65.70_{\pm 3.04}$ |
| GAN | $70.60_{\pm 3.41}$ | $48.80_{\pm 5.51}$ | $75.52_{\pm 4.96}$ | $69.98_{\pm 5.46}$ | $66.26_{\pm 3.72}$ |

*Table 6.* Ablation study on the density estimation scheme for learned GIN representations in GRATIN.

| Model | IMDB-BIN | IMDB-MUL | MUTAG | PROTEINS | DD |
|---|---|---|---|---|---|
| GMM w/ EM | $71.70_{\pm 4.24}$ | $49.20_{\pm 2.06}$ | $88.83_{\pm 5.02}$ | $71.33_{\pm 5.04}$ | $68.61_{\pm 4.62}$ |
| GMM w/ VBI | $71.40_{\pm 2.65}$ | $47.80_{\pm 2.22}$ | $88.30_{\pm 5.19}$ | $70.25_{\pm 4.65}$ | $67.82_{\pm 4.96}$ |
| KDE | $69.10_{\pm 3.93}$ | $41.46_{\pm 3.02}$ | $77.60_{\pm 6.83}$ | $60.37_{\pm 3.04}$ | $67.48_{\pm 6.18}$ |
| Copula | $70.60_{\pm 2.61}$ | $47.60_{\pm 2.29}$ | $88.30_{\pm 5.19}$ | $70.16_{\pm 4.55}$ | $67.91_{\pm 4.90}$ |
| GAN | $70.50_{\pm 3.80}$ | $48.40_{\pm 1.71}$ | $88.83_{\pm 5.02}$ | $71.33_{\pm 5.55}$ | $67.74_{\pm 4.82}$ |

We compare these approaches for both the GCN and GIN models in Tables 5 and 6, respectively. As noticed, GMM with EM consistently outperforms the alternative methods across most datasets in terms of accuracy. The VBI method, an alternative approach for estimating GMM parameters, yields comparable performance to the EM algorithm. This consistency across datasets highlights the effectiveness and robustness of GMMs in capturing the underlying data distribution.

In certain cases, particularly with the GIN model, we observed competitive performance from the GAN approach, which, unlike GMM, requires additional training. Hence, GMMs provide a more straightforward and efficient solution.

## F. Training and Augmentation time

We compare the data augmentation times of our approach and the baselines in Table 7. In addition to outperforming the baselines on most datasets, our approach offers an advantage in terms of time complexity. The training time of baseline models varies depending on the augmentation strategy used, specifically, whether it involves pairs or individual graphs. Even in cases where a graph augmentation has a low computational cost for some baselines, training can still be time-consuming as multiple augmented graphs are required to achieve satisfactory test accuracy. For instance, methods like DropEdge, DropNode, and SubMix, while computationally simple, require generating multiple augmented samples at each epoch,

thereby increasing the overall training time. Following the framework of (Yoo et al., 2022), we must sample several augmented graphs for each training graph at every epoch to achieve optimal results. In contrast, GRATIN introduces a more efficient approach by generating only one augmented graph per training instance, which is reused across all epochs. This design ensures a balance between computational efficiency and augmentation effectiveness, reducing the overall training burden while maintaining strong performance. The only baseline that is more time-efficient than our approach is GeoMix; however, our method consistently outperforms GeoMix across all settings, as shown in Tables 1 and 2.

*Table 7.* Mean training and augmentation time in seconds of our model in comparison to the other benchmarks.

|   | Time | Model | IMDB-BIN | MUTAG | DD |
|---|------|-------|----------|-------|-----|
| ① | Aug. Time | Vanilla | - | - | - |
|   |  | DropEdge | 0.02 | 0.01 | 0.01 |
|   |  | DropNode | 0.01 | 0.02 | 0.01 |
|   |  | SubMix | 1.27 | 0.23 | 0.45 |
|   |  | $\mathcal{G}$-Mixup | 0.74 | 0.11 | 4.26 |
|   |  | GeoMix | 2,344.12 | 73.52 | 1,005.35 |
|   |  | GRATIN | 2.87 | 0.51 | 3.25 |
| ② | Train. Time | Vanilla | 765.96 | 99.32 | 428.10 |
|   |  | DropEdge | 892.14 | 596.82 | 3,037.30 |
|   |  | DropNode | 884.71 | 803.63 | 3,325 |
|   |  | SubMix | 1,711.01 | 1,487.03 | 2,751.92 |
|   |  | $\mathcal{G}$-Mixup | 148.71 | 28.14 | 177.55 |
|   |  | GeoMix | 89.01 | 101.82 | 123.41 |
|   |  | GRATIN | 774.47 | 101.56 | 438.39 |

*Table 8.* Classification accuracy ($\pm$ std) on different benchmark datasets for the data augmentation baselines using the GCN backbone. Higher accuracy (in %) is better.

| Model | IMDB-BIN | IMDB-MUL | MUTAG | PROTEINS | DD |
|-------|----------|----------|-------|----------|-----|
| DropEdge | $71.40_{\pm 3.69}$ | $48.47_{\pm 2.36}$ | $73.88_{\pm 7.98}$ | $67.56_{\pm 4.50}$ | $66.04_{\pm 4.35}$ |
| DropNode | $71.80_{\pm 4.11}$ | $48.60_{\pm 3.45}$ | $72.78_{\pm 8.01}$ | $67.83_{\pm 4.75}$ | $66.55_{\pm 3.97}$ |
| SubMix | $72.50_{\pm 3.90}$ | $47.93_{\pm 3.58}$ | $71.72_{\pm 10.59}$ | $59.74_{\pm 2.83}$ | $62.25_{\pm 3.29}$ |

*Table 9.* Classification accuracy ($\pm$ std) on different benchmark datasets for the data augmentation baselines using the GIN backbone. Higher accuracy (in %) is better.

| Model | IMDB-BIN | IMDB-MUL | MUTAG | PROTEINS | DD |
|-------|----------|----------|-------|----------|-----|
| DropEdge | $71.70_{\pm 3.25}$ | $43.67_{\pm 6.58}$ | $72.36_{\pm 10.40}$ | $66.31_{\pm 5.49}$ | $70.20_{\pm 2.85}$ |
| DropNode | $71.00_{\pm 4.49}$ | $42.67_{\pm 5.54}$ | $75.00_{\pm 6.67}$ | $64.60_{\pm 4.52}$ | $70.80_{\pm 3.31}$ |
| SubMix | $71.10_{\pm 4.15}$ | $42.80_{\pm 3.93}$ | $84.53_{\pm 6.66}$ | $60.10_{\pm 3.78}$ | $70.03_{\pm 2.61}$ |

Furthermore, the performance of simple augmentation baselines such as DropEdge, DropNode, and SubMix significantly drops when using *only one augmentation* per training graph for all the epochs, which is the framework adopted in GRATIN, as shown in Tables 8 and 9. Tables 8 and 9, these baselines experience a significant drop in accuracy under this constraint, emphasizing their reliance on generating diverse augmentations at each epoch to maintain strong performance, c.f. the results in Tables 1 and 2.

The only graph augmentation baseline with comparable or better time complexity than GRATIN is $\mathcal{G}$-Mixup. However, GRATIN consistently outperforms $\mathcal{G}$-Mixup in most cases, as shown by the results in Tables 1 and 2.

# G. Augmentation Strategies

We present several graph data augmentation strategies, each defined by the augmentation function $A_\lambda$. For simplicity, we denote it $A_\lambda(\mathcal{G}_n)$ instead of the more explicit $A_\lambda(\mathcal{G}_n, y_n)$.

**Edge Perturbation.** Randomly adding or removing edges. Given a graph $\mathcal{G}_n = (\mathcal{V}_n, \mathcal{E}_n, \mathbf{X}_n)$, edge perturbation is defined as $A_\lambda(\mathcal{G}_n) = (\mathcal{V}_n, \mathcal{E}_n \cup \mathcal{E}_{\text{add}} \setminus \mathcal{E}_{\text{remove}}, \mathbf{X}_n)$, where edges in $\mathcal{E}_{\text{add}}$ and $\mathcal{E}_{\text{remove}}$ are sampled from a Bernoulli distribution $\mathcal{P}(\lambda) = \mathcal{B}(\lambda)$ with probability $\lambda$.

**Node Feature Perturbation.** Augmenting node features by introducing noise or masking some features. This is given by $A_\lambda(\mathcal{G}_n) = (\mathcal{V}_n, \mathcal{E}_n, \mathbf{X}_n + \lambda \mathbf{Z})$, where $\mathbf{Z} \sim \mathcal{N}(0, I)$ for Gaussian noise addition.

**Subgraph Sampling.** Extracting subgraphs. A common approach is $k$-hop neighborhood sampling, where $A_\lambda(\mathcal{G}_n) = (\mathcal{V}'_n, \mathcal{E}'_n, \mathbf{X}'_n)$, with $\mathcal{V}'_n \subseteq \mathcal{V}_n$ and $\mathcal{E}'_n$ being edges induced by the $k$-hop neighbors of randomly selected nodes selected based on a prior distribution $\mathcal{P}(\lambda)$.

**GRATIN.** In our approach, the augmented hidden representations $\mathbf{h}_{\widetilde{\mathcal{G}}} = A_\lambda(\mathbf{h}_\mathcal{G})$ corresponds to a sampled vector from the GMM distribution $\mathcal{P}_c$ that was previously fit on the hidden representations $\mathcal{H}_c = \{\mathbf{h}_\mathcal{G} \mid y_n = c\}$ of the graphs in the training set with the same class $c$. Formally, $\mathbf{h}_{\widetilde{\mathcal{G}}} = A_{\lambda_c}(\{\mathcal{H}_c \mid \mathcal{G} \in c\})$, where $\lambda_c$ are the parameter of the GMM distribution $\mathcal{P}_c$.

It is important to note that in some augmentation strategies, the augmented graphs $\widetilde{\mathcal{G}}_n^m$ may not explicitly depend on the specific training graph $\mathcal{G}_n$. Instead, they may be sampled or generated based on other factors, such as a general graph distribution or global augmentation rules. This flexibility allows the augmentation framework to capture a broader range of variations while maintaining consistency with the original training data.

# H. Graph Distance Metrics

Let us consider the graph structure space $(\mathbb{A}, \|\cdot\|_\mathbb{A})$ and the feature space $(\mathbb{X}, \|\cdot\|_\mathbb{X})$, where $\|\cdot\|_\mathbb{G}$ and $\|\cdot\|_\mathbb{X}$ denote the norms applied to the graph structure and features, respectively. When considering only structural changes with fixed node features, the distance between two graphs $\widetilde{\mathcal{G}}, \mathcal{G}$ is defined as

$$\left\| \widetilde{\mathcal{G}} - \mathcal{G} \right\| = \|\mathbf{A} - \widetilde{\mathbf{A}}\|_\mathbb{G}, \tag{4}$$

where $\widetilde{\mathbf{A}}, \mathbf{A}$ are respectively the adjacency matrix of $\widetilde{\mathcal{G}}, \mathcal{G}$, and the norm $\|\cdot\|_\mathbb{G}$ can be for example the Frobenius or spectral norm. If both structural and feature changes are considered, the distance extends to:

$$\left\| \widetilde{\mathcal{G}} - \mathcal{G} \right\| = \alpha\|\mathbf{A} - \widetilde{\mathbf{A}}\|_\mathbb{A} + \beta\|\mathbf{X} - \widetilde{\mathbf{X}}\|_\mathbb{X}, \tag{5}$$

where $\widetilde{\mathbf{X}}, \mathbf{X}$ are the node feature matrices of $\widetilde{\mathcal{G}}, \mathcal{G}$ respectively, and $\alpha, \beta$ are hyperparameters controlling the contribution of structural and feature differences.

In most baseline graph augmentation techniques, such as $\mathcal{G}$-Mixup, SubMix, and DropNode, the alignment between nodes in the original graph $\mathcal{G}$ and the augmented graph $\widetilde{\mathcal{G}}$ is known. However, in cases where the node alignment is unknown, we must take into account node permutations. The distance between the two graphs is then defined as

$$\left\| \widetilde{\mathcal{G}} - \mathcal{G} \right\| = \min_{P \in \Pi} \left( \alpha\|\mathbf{A} - P\widetilde{\mathbf{A}}P^T\|_\mathbb{A} + \beta\|\mathbf{X} - P\widetilde{\mathbf{X}}\|_\mathbb{X} \right), \tag{6}$$

where $\Pi$ is the set of permutation matrices. The matrix $P$ corresponds to a permutation matrix used to order nodes from different graphs. By using Optimal Transport, we find the minimum distance over the set of permutation matrices, which corresponds to the optimal matching between nodes in the two graphs. This formulation represents the general case of graph distance, which has been used in the literature (Abbahaddou et al., 2024).

A common choice for measuring the distance between two graphs is the norm applied to their adjacency matrices. One widely used norm is the Frobenius norm, defined as $\|\mathbf{A} - \widetilde{\mathbf{A}}\|_F = \sqrt{\sum_{i,j}(A_{ij} - \widetilde{A}_{ij})^2}$, which captures element-wise differences between the adjacency matrices. Another commonly used norm is the spectral norm, defined as $\|\mathbf{A} - \widetilde{\mathbf{A}}\|_2 = \sigma_{\max}(\mathbf{A} - \widetilde{\mathbf{A}})$, where $\sigma_{\max}$ denotes the largest singular value of the difference matrix.

## I. Gaussian Mixture Models

GMMs are probabilistic models used for modeling complex data by representing them as a mixture of multiple Gaussian distributions. The probability density function $p(\mathbf{x})$ of a data point $\mathbf{x}$ in a GMM with $K$ Gaussian components is given by:

$$p(\mathbf{x}) := \sum_{k=1}^{K} \pi_k \mathcal{N}(\mathbf{x} \mid \boldsymbol{\mu}_k, \boldsymbol{\Sigma}_k), \tag{7}$$

where $\pi_k$ is the weight of the $k$-th Gaussian component, with $\pi_k \geq 0$ and $\sum_{k=1}^{K} \pi_k = 1$, and $\mathcal{N}(\mathbf{x} \mid \boldsymbol{\mu}_k, \boldsymbol{\Sigma}_k)$ is the Gaussian probability density function for the $k$-th component, defined as,

$$\mathcal{N}(\mathbf{x} \mid \boldsymbol{\mu}_k, \boldsymbol{\Sigma}_k) := \frac{1}{(2\pi)^{d/2} \det(\boldsymbol{\Sigma}_k)^{1/2}} \times \exp\left(-\frac{1}{2}(\mathbf{x} - \boldsymbol{\mu}_k)^{\top} \boldsymbol{\Sigma}_k^{-1}(\mathbf{x} - \boldsymbol{\mu}_k)\right),$$

where $\boldsymbol{\mu}_k$ and $\boldsymbol{\Sigma}_k$ are respectively the mean vectors and the covariance vectors of the $k$-th Gaussian component, and $d$ the dimensionality of $\mathbf{x}$. The parameters of a GMM are typically estimated using the EM algorithm (Dempster et al., 1977), which alternates between estimating the membership probabilities of data points for each Gaussian component (Expectation step) and updating the parameters of the Gaussian distributions (Maximization step). GMMs are a powerful tool in statistics and machine learning and are used for various purposes, including clustering and density estimation (Ozertem & Erdogmus, 2011; Naim & Gildea, 2012; Zhang et al., 2021).

## J. Experimental Setup

In this section, we detail the experimental setup for the conducted experiments. The necessary code to reproduce all our experiments is available on github at: https://github.com/abbahaddou/GRATIN

**Datasets.** We evaluate our model on five widely used datasets from the GNN literature, specifically IMDB-BIN, IMDB-MUL, PROTEINS, MUTAG, and DD, all sourced from the TUD Benchmark (Morris et al., 2020). These datasets consist of either molecular or social graphs. Detailed statistics for each dataset are provided in Table 10. We split the dataset into train/test/validation set by 80%/10%/10% and use 10-fold cross-validation for evaluation following the recent work of Zeng et al. (2024). If a dataset does not contain node features, we follow the standard practice in GNN literature by using one-hot encoding of node degrees as input features.

*Table 10.* Statistics of the graph classification datasets used in our experiments.

| Dataset | #Graphs | #Features | Avg. Nodes | Avg. Edges | #Classes |
|---|---|---|---|---|---|
| IMDB-BIN | 1,000 | - | 19.77 | 96.53 | 2 |
| IMDB-MUL | 1,500 | - | 13.00 | 65.94 | 3 |
| MUTAG | 188 | 7 | 17.93 | 19.79 | 2 |
| PROTEINS | 1,113 | 3 | 39.06 | 72.82 | 2 |
| DD | 1,178 | 82 | 284.32 | 715.66 | 2 |
| COLLAB | 5,000 | - | 74.5 | 4914.4 | 3 |
| REDDIT-M5K | 4,999 | - | 508.5 | 1189.7 | 5 |

**Baselines.** We benchmark the performance of our approach against the state-of-the-art graph data augmentation strategies. In particular, we consider the DropNode (You et al., 2020), DropEdge (Rong et al., 2020), SubMix (Yoo et al., 2022), $\mathcal{G}$-Mixup (Han et al., 2022) and GeoMix (Zeng et al., 2024). For the robustness to structure corruption experiment, c.f. Table 3, we also included NoisyGNN (Ennadir et al., 2024), a recent method explicitly designed to enhance GNN robustness, as an additional baseline.

**Implementation Details.** We used the PyTorch Geometric (PyG) open-source library, licensed under MIT (Fey & Lenssen, 2019). The experiments were conducted on an RTX A6000 GPU. For the datasets from the TUD Benchmark, we used a size base split. We utilized two GNN architectures, GIN and GCN, both consisting of two layers with a hidden dimension of 32. The GNN was trained on graph classification tasks for 300 epochs with a learning rate of $10^{-2}$ using the Adam optimizer (Kingma & Ba, 2014). To model the graph representations of each class, we fit a GMM using the EM algorithm, running for 100 iterations or until the average lower bound gain dropped below $10^{-3}$. The number of Gaussians used in the GMM is

provided in Table 11. After generating new graph representations from each GMM, we fine-tuned the post-readout function for 100 epochs, maintaining the same learning rate of $10^{-2}$.

**Computation of Influence Scores.** Computing and inverting the Hessian matrix of the empirical risk is computationally expensive, with a complexity of $\mathcal{O}(N \times p^2 + p^3)$, where $p = |\theta|$ is the number of parameters in the GNN. To mitigate the cost of explicitly calculating the Hessian matrix, we employ implicit Hessian-vector products (iHVPs), following the approach outlined in Koh & Liang (2017).

*Table 11.* The optimal number of Gaussian distributions in the GMM for each pair of dataset and GNN backbone.

| Model | IMDB-BIN | IMDB-MUL | MUTAG | PROTEINS | DD |
|-------|----------|----------|-------|----------|----|
| GCN   | 40       | 50       | 10    | 10       | 2  |
| GIN   | 50       | 5        | 2     | 2        | 50 |

## K. Experiments on Large Datasets

We conducted additional experiments on larger-scale datasets, including COLLAB and REDDIT-MULTI-5K (Morris et al., 2020). The resufffor and GIN backbones are included in Table 12 and 13, which further confirm the effectiveness of our approach on graphs with a larger number of nodes and edges.

*Table 12.* Classification accuracy ($\pm$ std) on large datasets for the data augmentation baselines using the GCN backbone. Higher accuracy (in %) is better. The symbol '–' denotes instances where the method has an excessive augmentation time, i.e., exceeding 2 hours

|                | Vanilla          | DropEdge         | DropNode         | SubMix           | $\mathcal{G}-$Mixup | GeoMix           | GRATIN           |
|----------------|------------------|------------------|------------------|------------------|---------------------|------------------|------------------|
| COLLAB         | $79.94 \pm 1.61$ | $79.70 \pm 1.10$ | $79.62 \pm 1.84$ | $81.86 \pm 1.62$ | $81.76 \pm 1.58$    | $80.74 \pm 1.89$ | $\mathbf{82.28 \pm 1.82}$ |
| REDDIT-MULTI-5K | $48.88 \pm 2.31$ | $48.87 \pm 1.99$ | $48.73 \pm 2.39$ | $48.77 \pm 2.01$ | $46.23 \pm 2.74$    | –                | $\mathbf{49.31 \pm 1.56}$ |

*Table 13.* Classification accuracy ($\pm$ std) on large datasets for the data augmentation baselines using the GIN backbone. Higher accuracy (in %) is better. The symbol '–' denotes instances where the method has an excessive augmentation time, i.e., exceeding 2 hours

|                | Vanilla          | DropEdge         | DropNode         | SubMix           | $\mathcal{G}-$Mixup | GeoMix           | GRATIN           |
|----------------|------------------|------------------|------------------|------------------|---------------------|------------------|------------------|
| COLLAB         | $77.80 \pm 1.53$ | $78.26 \pm 1.46$ | $78.86 \pm 2.09$ | $80.98 \pm 1.24$ | $78.89 \pm 2.33$    | $78.20 \pm 1.31$ | $\mathbf{79.08 \pm 1.13}$ |
| REDDIT-MULTI-5K | $51.85 \pm 4.29$ | $44.52 \pm 9.58$ | $50.87 \pm 3.36$ | $49.93 \pm 3.63$ | $50.63 \pm 4.04$    | –                | $\mathbf{51.53 \pm 3.54}$ |

## L. Softmax Confidence and Entropy Distributions

One of the critical challenges in training GNN is Softmax saturation, where the model produces confident predictions. This high confidence leads to vanishing gradients, making the influence score of the augmented graphs converge to zero. To analyze this phenomenon, we examine the Softmax confidence and entropy distributions for GCN and GIN across the different graph classification datasets. In Figure 4, 5, 6, 7 and 8, we present histograms illustrating the distribution of maximum Softmax confidence and entropy for models trained on the original DD, IMDB-BIN, IMDB-MUL, MUTAG, and PROTEINS datasets. Each dataset panel contains two histograms, the Confidence Histogram and the Entropy Distribution. The *Confidence Histogram* illustrates the distribution of the maximum Softmax confidence scores assigned by the model to the predicted class. A higher confidence value indicates that the model is more certain about its classification decision. The *Entropy Distribution* provides a measure of uncertainty in the model's predictions, computed as: $H(y) = -\sum_c y_c \log(y_c)$, where, for each class $c$, $y_c$ is the predicted probability corresponding to this class. Lower entropy values reflect high-confidence predictions and higher entropy values indicate more significant uncertainty.

Specifically, in the DD dataset, we observe an extreme case of Softmax saturation in GIN, where almost all predictions collapse to near-maximum confidence. The entropy histogram further reinforces the Softmax saturation in GIN on DD, where entropy values are heavily skewed towards zero, meaning the model rarely assigns significant probability mass outside the predicted class. Compared to other dataset-model settings, this effect is particularly pronounced in GIN trained on DD. In contrast, GCN exhibits a wider confidence distribution, maintaining a more balanced uncertainty.

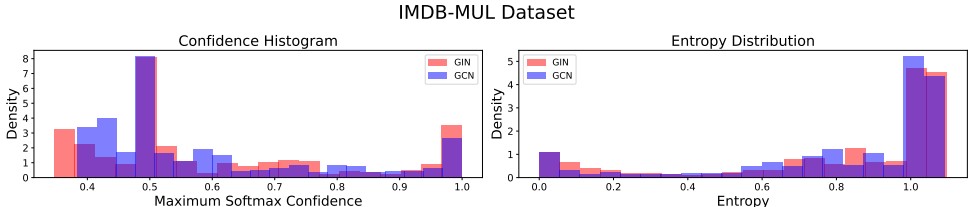

*Figure 4.* Softmax confidence and entropy distributions for the IMDB-BIN dataset.

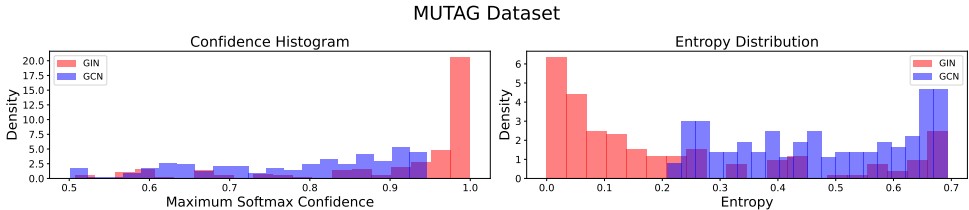

*Figure 5.* Softmax confidence and entropy distributions for the IMDB-MUL dataset.

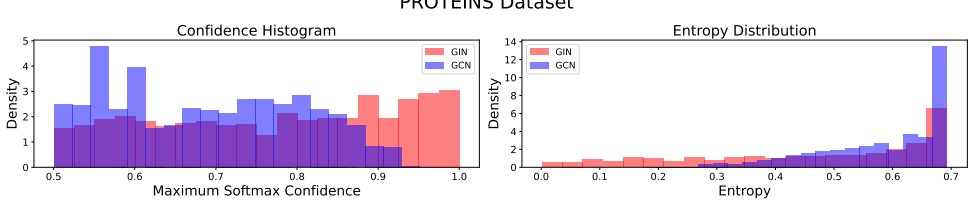

*Figure 6.* Softmax confidence and entropy distributions for the MUTAG dataset.

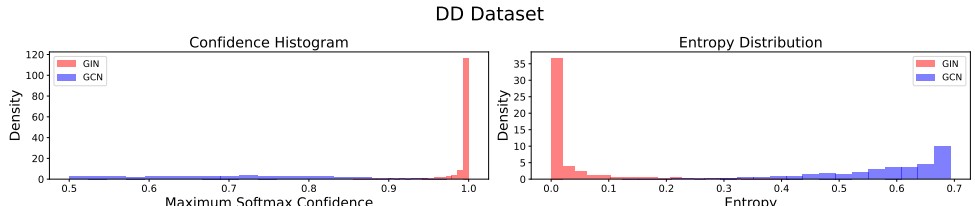

*Figure 7.* Softmax confidence and entropy distributions for the PROTEINS dataset.

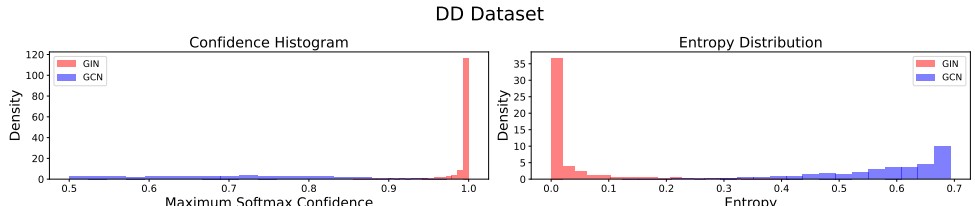

*Figure 8.* Softmax confidence and entropy distributions for the DD dataset.

