# OpenReview forum: "Graph Neural Network Generalization With Gaussian Mixture Model Based Augmentation"
_ICML.cc/2025/Conference — ICML 2025 poster_

### Official Review · Reviewer_wgAh · 2025-03-11

**Overall Recommendation:** 3

**Summary:**

This paper introduces GRATIN, a novel graph data augmentation method using Gaussian Mixture Models (GMMs) to enhance the generalization of Graph Neural Networks (GNNs) for graph classification. It provides a theoretical framework analyzing the impact of augmentation on GNN generalization via Rademacher complexity, designs the GRATIN algorithm to generate augmented data in the hidden representation space, and demonstrates superior performance in generalization and robustness compared to existing methods through experiments on multiple benchmark datasets.

#Update after rebuttal

**Claims And Evidence:**

This paper introduces GRATIN, a novel graph data augmentation method using Gaussian Mixture Models (GMMs) to enhance the generalization of Graph Neural Networks (GNNs) for graph classification. It provides a theoretical framework analyzing the impact of augmentation on GNN generalization via Rademacher complexity, designs the GRATIN algorithm to generate augmented data in the hidden representation space, and demonstrates superior performance in generalization and robustness compared to existing methods through experiments on multiple benchmark datasets.

**Essential References Not Discussed:**

No.

**Experimental Designs Or Analyses:**

Yes. I have check the experimental designs.

**Methods And Evaluation Criteria:**

The proposed methods and evaluation criteria are suitable for the problem.

**Other Comments Or Suggestions:**

1) Key formulas should be numbered to better align with the steps in Algorithm 1.

**Other Strengths And Weaknesses:**

**Strengths**

1) The method shows strong performance across multiple datasets,


**Weaknesses**

1) The organization of Section 3 is somewhat confusing and detracts from readability.
2) The relationship between the model and the Graph needs further clarification, particularly in distinguishing between topology and attribute distribution shifts.

**Questions For Authors:**

1) What are the differences between OOD in graph classification and node classification tasks?

2) What are the differences between the method described in Line 229 and MMD?

**Relation To Broader Scientific Literature:**

GRATIN's contributions are closely related to existing research on graph data augmentation and GNN generalization.

**Theoretical Claims:**

The theoretical proofs (e.g., Theorem 3.1 and Proposition 3.2) are sound and well-presented, offering new insights into the impact of data augmentation on GNN generalization.

---

> ### Author Rebuttal · Authors · 2025-03-31
>
> Response to Reviewer wgAh
> ==============
>
> We thank Reviewer wgAh for the feedback. In what follows, we address the raised questions and weaknesses point-by-point.
>
> ***[Weakness 1] Section 3 Clarity*** We appreciate the feedback and will do our best to improve the organization and clarity of Section 3 in the camera-ready version.
>
> ***[Weakness 2] Clarifying Topology vs. Attributes*** A topology shift refers to changes in the structure of the graph, such as the addition or deletion of edges, modifications in connectivity patterns, or global properties like graph density. On the other hand, attribute shift refers to changes in node features, such as different feature distributions and scales. Since GRATIN operates in the hidden space learned using several message passing neural network layers, it naturally captures both topological and attribute variations, leading to robust performance against diverse types of shifts. We will further clarify this advantage in the camera-ready version.
>
> ***[Question 1] OOD Differences: Graph vs. Node Classification*** The primary distinction between OOD in graph and node classification lies in the level at which distribution shifts occur. In graph classification, each sample is an entire graph, and the task is typically approached as a fully supervised problem, with a label for every training graph. Consequently, OOD shifts tend to manifest as substantial changes in global structure or attributes. For example, training graphs may be relatively small or simple, while test graphs might be larger or more complex. Conversely, in node classification, each sample is an individual node within a large graph, and the problem is addressed via semi-supervised learning, where only a subset of nodes have labels and the rest must be inferred from the graph’s topology and feature structure. Here, OOD shifts can arise locally, such as when training nodes appear in sparse neighborhoods while test nodes appear in denser regions or when training features lie within a certain range and test features experience domain drift. These differences require distinct OOD modeling strategies: node-level tasks, which often rely on partial labels and independent features, must manage  semi-supervised constraints, whereas graph-level tasks, typically with full supervision on entire graphs, emphasize broader structural or attribute shifts across distinct graphs. Moreover, unlike most baseline approaches, our method, GRATIN, is flexible and can be extended to node classification tasks. We have presented the details of this extension, including experimental results, in our response to Reviewer eSPE (c.f. [Question 4]), and we will incorporate this discussion into the camera-ready version of the manuscript.
>
>
> ***[Question 2] Line 229 vs. MMD*** The term in Line 229 is the expected distance between a representation of an original graph $\mathbf{h}$ and an augmented representation $\tilde{\mathbf{h}}$. By contrast, Maximum Mean Discrepancy (MMD) aligns full distributions by comparing mean embeddings in a Reproducing Kernel Hilbert Space (RKHS). While both aim to reduce the gap between original and augmented data, MMD enforces a global distribution-level alignment, whereas Line 229 focuses on local instance-level consistency. Despite these differing scopes, they share the overarching goal of encouraging similarity between real and augmented samples. Practically, the term in Line 229 is simpler, computationally cheaper, and better suited for theoretically motivating our data augmentation strategy.
>
>
> ***[Suggestion 1] Formula Numbering*** We thank the reviewer for the suggestion. In the current version, we chose to number only the equations that are explicitly referenced in the text. However, we agree that numbering all equations would improve clarity and alignment with Algorithm 1. We will revise the manuscript accordingly and include equation numbers in the camera-ready version to enhance readability.

---

> > ### Comment · Reviewer_wgAh · 2025-04-08
> >
> > Thank you for your reply.
> > I understand that variations in the hidden space can unify both topological and attribute variations. However, wouldn't directly perturbing the features to achieve augmentation be a simpler approach under the control of perturbation magnitude? This method could also simulate any kind of perturbation. So, can the author theoretically or experimentally explain what the key role of the Gaussian mixture model is?

---

> > > ### Author Response · Authors · 2025-04-09
> > >
> > > We thank Reviewer wgAh for their follow-up questions.
> > >
> > > **1.Feature Perturbation.** Adding noise to raw features may seem simpler, but overlooks the coupling between topology and node attributes.
> > >
> > > First, recall that after a graph $(\mathbf{A}, \mathbf{X})$ passes through a GNN to produce a graph-level embedding $\mathbf{h_G}$ capturing both the node features and the graph’s topology. If we only perturb the features while keeping the adjacency matrix fixed, we obtain an embedding distribution denoted by $\mathcal{D_{\text{feat-only}}}=\lbrace\mathbf{h_{\tilde{G}}}| \tilde{G}=(\mathbf{A},\mathbf{X}+\delta\mathbf{X})\rbrace,$
> > > where $\delta \mathbf{X}$ is drawn from a distribution. Similarly, if we perturb the graph stucture only, we obtain another distribution $\mathcal{D_{\text{struct-only}}}=\lbrace \mathbf{h_{\tilde{G}}}|\tilde{G}=(\mathbf{A}+\delta \mathbf{A}, \mathbf{X})\rbrace.$ In contrast, jointly perturbing both features and structure yields a third distribution $\mathcal{D_{\text{combined}}}=\lbrace \mathbf{h_{\tilde{G}}}|\tilde{G}=(\mathbf{A}+\delta \mathbf{A},\mathbf{X}+\delta \mathbf{X})\rbrace,$ which generalizes the other two. Since the GNN’s embedding depends nonlinearly on both $\mathbf{A}$ and $\mathbf{X}$, we have $\mathcal{D_{\text{feat-only}}} \neq \mathcal{D_{\text{struct-only}}} \neq \mathcal{D_{\text{combined}}},$ and each covers a different subspace of the learned representation manifold.
> > >
> > > To understand this, consider a bottleneck structure in a graph, such as a single edge acting as a bridge between two otherwise disconnected components. Perturbing the node’s features might affect its representation locally, but perturbing the structure, e.g., removing that connecting edge, has a significantly greater impact on the message-passing dynamics, potentially altering how information flows across the entire graph. Similarly, in sparse graphs, including most real-world datasets, this problem becomes more pronounced, as many nodes already have limited connections, and feature perturbations alone cannot introduce the topological changes required to mimic realistic scenarios.
> > >
> > > Nevertheless, we acknowledge the suggested augmenation strategy, so we added an experiment applying Gaussian noise of varying magnitudes, i.e., we sampled $\delta\mathbf{X}\sim\mathcal{N}(0,\sigma^2 I)$ with different values of $\sigma$, and perturbed the input features as $\tilde{\mathbf{X}}=\mathbf{X}+\delta \mathbf{X}$, keeping $\mathbf{A}$ unchanged. As shown in the table below, GRATIN consistently outperformed this approach, highlighting its ability to generate meaningful augmentations by capturing variations in both topology and features.
> > >
> > > |Method|IMDB-MUL|MUTAG|PROTEINS|
> > > |-|-|-|-|
> > > |Feat. only (0.01)|47.06 ±2.21|74.97± 7.71|70.16 ±4.89|
> > > |Feat. only (0.05)|46.20 ±4.42|73.86 ±7.21|70.07± 5.12 |
> > > |Feat. only (0.1)|46.26±4.37|75.49±7.78|69.36±5.16|
> > > |GRATIN|**49.82±4.26**|**76.05±6.74**|**70.97±5.07**|
> > >
> > > **2. The Role of the Gaussian Mixture Model (GMM)**
> > >
> > > ***2.1.Theory.*** Using GMMs is supported by the theory in our paper. Thm 3.1 shows that the generalization gap can be bounded by the Rademacher complexity, which itself depends on how close the augmented data distribution is to the real one. This provides a theoretical basis for ensuring that data augmentation should not introduce arbitrary perturbations but rather ones that respect the geometry of the learned representation space. Additionally, Thm 3.3 establishes that GMMs are universal density approximators. This guarantees that the GMM can faithfully approximate the true distribution of graph representations. Finally, Proposition 3.2 refines this idea by providing a bound on the expected perturbation between the original and augmented data. Specifically, it links the deviation to two terms: the KL divergence between the real and augmented distributions and the supremum distance in representation space. By fitting a GMM to the real data, we can reduce the KL divergence. Moreover, due to the exponential decay of Gaussian distributions, the supremum distance $\sup_{\mathbf{h}\sim\delta_\mathcal{D},  \tilde{\mathbf{h}}\sim Q_\lambda}\|\mathbf{h}-\tilde{\mathbf{h}}\|$ is naturally constrained, ensuring a better control of the expected distance $\mathbb{E}_{\mathbf{h}\sim\delta_D, \tilde{\mathbf{h}}\sim Q}[\|\mathbf{h}-\tilde{\mathbf{h}}\|]$.
> > >
> > > ***2.2.Empirics.***  Empirically, we chose GMMs also for their efficiency. Unlike generative models such as GANs or VAEs, GMMs are fast to fit and require no adversarial training or reconstruction objectives. This efficiency suits GNNs. Sampling from mixture components yields diverse, coherent augmentations. As shown in Appendix F (ablation study), we compared GMM based sampling with alternative strategies. The GMM consistently outperformed these baselines, particularly in terms of generalization ability.
> > >
> > > Thus, GMMs offer both theoretical and practical benefits.

---

### Official Review · Reviewer_eSPE · 2025-03-13

**Overall Recommendation:** 4

**Summary:**

The authors propose a novel graph data augmentation method, GRAFIN, which leverages Gaussian Mixture Models (GMMs) to learn the distribution of hidden representations generated by a trained Graph Neural Network (GNN). The method then augments the training data based on this learned distribution. Furthermore, the authors provide theoretical results that support the correctness of the proposed approach.

**Claims And Evidence:**

Yes

**Essential References Not Discussed:**

No

**Experimental Designs Or Analyses:**

Yes, all experiments in the main text seem correct. The obtained results in Table 3 are bolded as the best, but there are no statistically significant differences compared to the other methods.

**Methods And Evaluation Criteria:**

Yes

**Other Comments Or Suggestions:**

Comments / Suggestions:
1. Page 3: Replace “Augemtation” with “Augmentation.”
2. Figure 1: The steps presented in the figure’s caption should be marked on the figure for clarity.
3. Page 16: Correct “us to e perform” to “us to perform.”
4. Page 16 (redundant repetition): “The parameters θ̂ and θ̂” should be revised to avoid redundancy.
5. Structure space term clarification:
  - Original: “The findings of Theorem 3.1 hold for all norms defined on the graph input space. Specifically, let us consider the graph structure space [...]”
  - Suggested: The term “graph structure space” might be unclear for readers; consider providing a brief explanation or rephrasing for clarity.
6. Misleading notation clarification:
  - Original: “[…] where G denotes the space of all possible graphs […]”
  - Suggested: “G should denote the space of all possible graphs whose distribution matches that of the training/validation/test sets.”
7. Line 936: “nodes in the graph, d is hidden dimension” should likely be corrected to “nodes in the graph, where $d_t$ is the hidden dimension.”


  -

**Other Strengths And Weaknesses:**

Strengths:


1. The authors propose a novel method to augment graph data.
2. The paper is well-written and easy to follow.
3. The authors present a solid theoretical framework that supports their claims.
4. The proofs of the theorems are well-structured.

Weaknesses:

1. There are typos that need to be fixed to maintain the professionalism of the paper.
2. The proposed approach should be validated on datasets with a larger number of graphs and larger graphs in terms of nodes and edges, such as COLLAB and REDDIT-MULTI-5K.
3. Equations should be numbered.
4. In the proposed approach, the authors rely on the law of large numbers, which applies for sufficiently large m (notation from the paper), where m is the number of generated graphs per graph in the training set. I am not convinced that this reasoning holds when the number of generated graphs is much smaller.

**Questions For Authors:**

Questions:
1. Could you elaborate more on the application of the law of large numbers? Is it still applicable if the number of generated graphs (m) is smaller than one per graph in the training set?
2. Could you explain why, after fitting the GMM to representations produced by a GNN, only the learnable parameters of the post-readout function $\psi$ are fine-tuned? If the reason is that you do not want to modify the GNN weights responsible for producing hidden representations, I believe it is worth explicitly mentioning.
3. I do not understand the last four lines at the end of page 12. Could you please clarify what you mean by introducing indexing?
4. Can we extend your framework to node classification? Is that possible?

**Relation To Broader Scientific Literature:**

GMM-GDA builds on prior work in graph data augmentation (e.g., GAug and G-Mixup) by using Gaussian Mixture Models (GMMs) to generate synthetic graph data, improving GNN generalization. Unlike previous methods, it provides a theoretical framework using Rademacher complexity to bound generalization error. Its efficiency and theoretical grounding make it a significant step forward in enhancing GNNs with data augmentation.

**Theoretical Claims:**

Yes. I checked all of them (main text + supplementary materials) and found the following issues/suggestions:
1. In the proof of Theorem 3.1, the penultimate inequality of this proof can be replaced with an equality due to the vanishing of Rademacher variables under the absolute value.
2. In the proof of Theorem 3.1, in the final equation, shouldn't it be $G_n^{m}$​ instead of $G_n^{\lambda}$​?
3. It looks like there's a compilation issue related to mathopdh on page 15.
4. End of page 15: summation from lowercase n=1, not n=N.
5. Line 766: There appears to be a potential issue when the measures return the same value for distinct arguments, which could result in a division by zero.

---

> ### Author Rebuttal · Authors · 2025-03-31
>
> We thank Reviewer eSPE for their review. In what follows, we address the raised questions point-by-point.
>
> **[R1] Penultimate Inequality in the Proof of Thm 3.1**
> Thank you, indeed, the inequality can be replaced with an equality. We will update the proof in the camera-ready (CR) version.
>
> **[R2, R3, R4, W1] Typos** We are grateful to the reviewer for spotting the typos. We'll correct them in the CR version.
>
> **[R5] Division by Zero** In our framework, $\delta_{\mathcal{D}}$ is a discrete distribution, where each $h\sim\delta_{\mathcal{D}}$ has equal probability $1/N$. Consequently, we disregard sampling augmented representations $\tilde{h}$ such that $Q_\lambda(\tilde{h}) = 1/N$, which is highly unlikely to occur in practice. We'll add a technical assumption in Prop. 3.2 to ensure that our choice of $Q_\lambda$ excludes augmented representations $\tilde{h}$ for which $Q_\lambda(\tilde{h})=1/N$. This assumption formally avoids the risk of division by zero.
>
> **[W2] Large datasets** The requested results are included in Table 1 below and further confirm the effectiveness of GRATIN on graphs with a larger number of nodes and edges.
>
> Table 1: GRATIN for larger datasets. The symbol ‘--’ indicates augmentation time exceeding 2 hours.
>
> ||Model|Vanilla|DropEdge|DropNode|SubMix|G-Mixup|GeoMix|GRATIN|
> |-|-|-|-|-|-|-|-|-|
> |COLLAB|GCN|79.94±1.61|79.70±1.10|79.62±1.84|81.86±1.62|81.76±1.58|80.74±1.89|**82.28±1.82**|
> ||GIN|77.80±1.53|78.26±1.46|78.86±2.09|**80.98±1.24**|78.89±2.33|78.20±1.31|80.07±1.35|
> |REDDIT-M5K|GCN|48.88±2.31|48.87±1.99|48.73±2.39|48.77±2.01|46.23±2.74|--|**49.31±1.56**|
> ||GIN|51.85±4.29|44.52±9.58|50.87±3.36|49.93±3.63|50.63±4.04|--|**52.01±3.54**|
>
> **[W3] Formula Numbering** We chose to number only equations that are explicitly referenced in the text. However, we’ll number all equations in the revised manuscript to improve readability.
>
> **[Other Comments]** We appreciate the detailed suggestions. We'll align figure captions more closely with visual steps and rephrase ambiguous terms to improve clarity.
>
> **[Q1 and W4] Law of Large Numbers (LLN)** We invoke the LLN primarily for theoretical considerations, illustrating that our method’s assumptions hold as $m$ (the number of augmented graphs per original sample) grows large. In practice, however, even small $m$ consistently improve accuracy without incurring a substantial computational cost (see additional experiments varying $m$ in our response to Reviewer 4GM3, c.f. S3-Part 2). Our findings confirm that a modest number of augmentations can already be beneficial, indicating that while the LLN provides a formal theoretical foundation, our approach remains effective when $m$ is limited.
>
> **[Q2] Post-readout Fine-tuning** We split the GNN into two parts: (1) message-passing layers that produce graph-level representations and (2) a shallow post-readout function $\psi$ that maps these representations to final predictions. After training the message-passing layers on the classification task, we fit a GMM to the resulting graph-level representations. This hidden space becomes the manifold where we perform our augmentation. If we were to update the message-passing layers after fitting the GMM, the structure of the learned representation space would shift, making the distribution modeled by the GMM inconsistent and, thus, degrading the quality of the augmented samples. To avoid this, we keep the message-passing layers fixed and fine-tune only $\psi$, which is computationally efficient, scalable and capable of adapting to the augmented dataset. Retraining the message-passing layers would increase computational time. Importantly, the test time prediction remains a composition of the fixed GNN encoder and the updated post-readout function, meaning that the GNN weights responsible for producing hidden representations still play a central role in the model's predictions.
>
> **[Q3] McDiarmid’s Inequality** The line break may have introduced confusion. We refer to the standard application of McDiarmid’s inequality by considering two datasets that differ at exactly one index. This setup allows us to bound the change in the expected loss. We'll revise the text to make this explanation clearer in the final version.
>
> **[Q4] Node Classification** GRATIN can be extended for tasks like node classification. While other methods might also be adaptable, such extensions are not always straightforward in their original formulations. The extension follows the same framework but shifts focus to node-level distributions. We train a GNN, fit class-wise GMMs on node embeddings, sample new representations for augmentation, and retrain a shallow classifier. Experiments with the GCN on widely used node classification datasets (Table 2) demonstrate effectiveness beyond graph-level tasks.
>
> Table 2:GRATIN for node classification.
>
> ||Cora|CiteSeer|PubMed|CS|
> |-|-|-|-|-|
> |Vanilla|**80.71±0.61**|70.36±0.90|79.78±0.28|89.45±1.25|
> |GRATIN|80.47±0.51|**71.08±0.71**|**79.85±0.22**|**90.82±0.76**|

---

> > ### Comment · Reviewer_eSPE · 2025-04-01
> >
> > The authors answered all my concerns. I raised the score.

---

> > > ### Author Response · Authors · 2025-04-04
> > >
> > > Thank you for acknowledging our clarification and for raising your score. We greatly appreciate your constructive feedback.

---

### Official Review · Reviewer_4GM3 · 2025-03-24

**Overall Recommendation:** 3

**Summary:**

This paper introduces GRATIN, a novel graph data augmentation approach leveraging Gaussian Mixture Models (GMMs) to enhance the generalization and robustness of Graph Neural Networks (GNNs). The authors argue that GNNs often face challenges in generalizing to out-of-distribution (OOD) data, especially with limited or imbalanced datasets. The proposed method generates augmented graph representations by modeling the distribution of graph embeddings (via GMMs) in the hidden representation space. GRATIN is supported by a theoretical framework based on Rademacher complexity and influence functions, which quantifies the impact of augmentation on generalization performance. Extensive experiments on benchmark graph classification datasets demonstrate GRATIN's effectiveness, achieving competitive or superior results compared to existing augmentation methods.

**Claims And Evidence:**

I think the main claims in the paper are well supported.

**Essential References Not Discussed:**

I suggest discussing GMM-based augmentation for graph representation learning.

Such as:

[1] Li, Yanjin, Linchuan Xu, and Kenji Yamanishi. "GMMDA: Gaussian Mixture Modeling of Graph in Latent Space for Graph Data Augmentation." IEEE International Conference on Data Mining (ICDM), 2023.
[2] Fukushima, Shintaro, and Kenji Yamanishi. "Graph Community Augmentation with GMM-based Modeling in Latent Space." IEEE International Conference on Data Mining (ICDM), 2024.

**Experimental Designs Or Analyses:**

See Methods And Evaluation Criteria.

**Methods And Evaluation Criteria:**

Overall, the paper conducts reasonable evaluations.

It can be further enhanced:

1) The paper only studies structure perturbation. Is the model robust to feature perturbation?

2) It is suggested to include learning-based automated graph augmentations, such as:
[1] Luo, Youzhi, et al. "Automated Data Augmentations for Graph Classification." The Eleventh International Conference on Learning Representations (ICLR). 2023

3) sensitivity analysis of key hyperparameters, such as the number of Gaussian components in the GMM or the number of augmented samples.

**Other Comments Or Suggestions:**

N.A.

**Other Strengths And Weaknesses:**

Strengths:

1) The use of Gaussian Mixture Models (GMMs) for graph data augmentation at the hidden representation level is both innovative and computationally efficient. Unlike traditional augmentation methods that operate directly on graphs (e.g., DropNode, DropEdge, Mixup variants), GRATIN focuses on the latent space, which avoids costly node alignments and allows architecture-specific augmentations.

2) The paper provides a strong theoretical foundation for the proposed method. The use of Rademacher complexity to analyze generalization bounds and influence functions to measure the impact of augmented data on test performance demonstrates a deep understanding of the augmentation problem.

3) GRATIN is evaluated on a diverse set of graph classification datasets (e.g., IMDB-BIN, MUTAG, PROTEINS, DD) using two prominent GNN architectures (GCN and GIN). The results show that GRATIN achieves strong generalization performance and robustness to structural perturbations compared to baseline methods like SubMix, G-Mixup, and GeoMix.

**Questions For Authors:**

1) Is the model robust to feature perturbation?
2) Could the hyper-parameters introduced in the method be integrated into the augmentation process dynamically during training? Such as the augmented representations filtering proportion.

**Relation To Broader Scientific Literature:**

1) Advancing Mixup-Based Techniques: GRATIN contributes to the literature by leveraging Gaussian Mixture Models (GMMs) to generate augmented graphs in the hidden representation space rather than directly modifying graph structures. This makes the augmentation highly efficient.
2) Theoretical Focus: While most prior augmentation methods lack theoretical guarantees, GRATIN provides a rigorous analysis of generalization improvements through augmentation using influence functions and regret bounds.

**Theoretical Claims:**

I am not able to check the correctness of the theory, because I am not familiar with the model generalization theory.

---

> ### Author Rebuttal · Authors · 2025-03-31
>
> Response to Reviewer 4GM3
> ======
> We thank Reviewer 4GM3 very much for their careful review. In what follows, we answer point-by-point.
>
> **[S1 and Q1]  Robustness Exp.** While our primary contribution lies in improving generalization via data augmentation, we also address robustness as a secondary but significant aspect, building on the recent framework introduced by the baseline GeoMix. Since, in GRATIN we act on the hidden space generated by several GNN layers, our embedding space jointly considers the structure and features by design (unlike most baselines that focus only on structural perturbations). This comprehensive approach allows our model to significantly outperform baselines under feature perturbation, where others often fail due to their limited augmentation scope. We employed a standard feature perturbation baseline by injecting Gaussian noise $\mathcal{N}(0, I) $ into node features with a scaling parameter $\beta=0.5$. We compared GRATIN against a Vanilla GNN (no augmentation) and the recent GeoMix baseline. The results of this experiment, shown in Table 1, demonstrate that GRATIN achieves superior robustness to feature perturbations across both datasets.
>
> Table 1: Robustness to features perturbation
> |Dataset| Vanilla | GeoMix|GRATIN|
> |-|-|-|-|
> |PROTEINS|64.96± 4.08|64.87± 9.28|**69.26±3.52**|
> |MUTAG|62.77±12.40|65.40±8.28|**68.56± 8.81**|
>
> **[S2] Additional Baseline** We appreciate the reviewer’s suggestion. We ran additional experiments using the augmentation method proposed in Luo et al. (ICLR 2023), referred to as GraphAug. This model is applied to a GCN backbone following the same experimental setup used for GRATIN and the rest baselines used in our paper. The results, presented in Table 2, will be included in the camera-ready version of the manuscript. GRATIN outperforms GraphAug across multiple datasets, demonstrating the effectiveness of our augmentation strategy that simultaneously operates over both structure and feature spaces.
>
> Table 2: GraphAug Results
> | Method|IMDB-BIN|IMDB-MUL|MUTAG|PROTEINS|DD|
> |-|-|-|-|-|-|
> | GraphAug |73.91±4.62|49.66±4.22|73.36±9.30|70.80±3.89|71.02±3.84 |
>
> **[S3-Part 1] Sensitivity to GMM Components** We experimented with $K$ components in the GMM in the range from 2 to 50. Below, we include a hyperparameter sensitivity analysis conducted on the GIN backbone using the IMDB-BIN dataset, where we observed consistent results across different numbers of Gaussian components $K$.
>
> Table 3: Impact of $K$ (IMDB-BIN)
> |K|10|20|30|40|50|
> |-|-|-|-|-|-|
> |GRATIN-GIN|71.12±2.70|71.34±2.28|71.38±2.57|71.64±2.72|71.74±4.24|
>
> **[S3-Part 2] Sensitivity to the number of augmented samples**  We conducted a sensitivity analysis to study the effect of the number of augmented samples per graph $m$ on model performance. As shown in the table below, the performance of GRATIN remains stable across a wide range of augmentation levels. This indicates that our method is robust to the choice of this hyperparameter, with accuracy variations within a narrow range even when increasing the number of augmentations from 1 to 30.
>
> Table 4: Impact of $m$ (GRATIN-GCN)
> |$m$|1|5|10|20|30|
> |-|-|-|-|-|-|
> |DD|71.90±2.81|72.25±3.26|72.02±3.34|71.87±3.34|71.81±3.29|
> |MUTAG|76.05±6.74|75.53±6.76|75.53±6.76|75.37±6.54|75.16±6.42|
>
> **[S4] Related Work**  Thank you for pointing out these relevant works. We appreciate the suggestions and will include both references in the camera-ready version of the manuscript. GMMDA focuses on node classification, proposing a GMM-based augmentation that preserves labels through MDL-guided sampling of synthetic nodes. GCA, on the other hand, targets graph community augmentation, generating unseen graphs with new community structures by introducing new clusters in the latent space. Both works support the general motivation of modeling latent graph representations with GMMs, which aligns with our method. However, our approach differs in that it uses GMM sampling not to generate new nodes or community structures but to augment graph-level representations in the hidden space of a GNN. We view these works as complementary and helpful in motivating GMM-based augmentation in GNNs, and will cite them accordingly.
>
> **[Q2] Dynamic Integration of Hyperparameters** In our current setup, certain parameters, such as the number of Gaussian components, were observed to have a negligible impact, whereas others, like the number of augmentations, directly influence the size of the training set and, consequently, the overall training time. The filtering proportion for augmented representations, which we currently fix based on influence-based heuristics, is indeed a good candidate for a dynamic approach. In principle, we could integrate an attention mechanism or adapt the filtering as an Active Learning problem, allowing the model to automatically learn the most informative augmented samples. We see this as a promising future direction for further improving generalization.

---

### Decision · Program_Chairs · 2025-05-01

**Decision:**

Accept (poster)

**Comment:**

There is a general consensus among the reviewers to accept the paper.